# Prevalence of visual snow and relation to attentional absorption

**Rui Miguel Costa** [1]*, **Pedro Campos**[2], **Madalena Wiborg**[2], **Catarina Rebôlo**[2], **Marc Wittmann**[3], **Jürgen Kornmeier**[3,4]

1 William James Center for Research, Ispa–Instituto Universitário, Lisbon, Portugal, 2 Ispa–Instituto Universitário, Lisbon, Portugal, 3 Institute for Frontier Areas of Psychology and Mental Health, Freiburg, Germany, 4 Faculty of Medicine, Department of Psychiatry and Psychotherapy, Medical Center—University of Freiburg, Freiburg, Germany

* rcosta@ispa.pt

## Abstract

Visual snow is a condition of unclear prevalence characterized by tiny flickering dots throughout the entire visual field. It appears to result from visual cortex hyperactivity and possibly correlates with propensity to be engrossed in sensory and imaginary experiences (absorption). The prevalence and correlates of visual snow, and emotional reactions to it, were explored in the general Portuguese population with three studies with online surveys. In Study 1, 564 participants were shown an animated graphic simulation of visual snow and asked to rate how frequently they have similar percepts on a scale anchored by 0% and 100% of their waking time. They also reported their degree of distress and fascination resulting from visual snow. Absorption was measured with the Modified Tellegen Absorption Scale. 44% of respondents reported they see visual snow at least 10% of the time, and 20% reported seeing it between 80% and 100% of the time. Similar to findings in clinical samples, the frequency of visual snow correlated with tinnitus frequency and entoptic phenomena, but not with ophthalmologic problems. It was confirmed that visual snow is related to absorption. Although distress caused by visual snow was generally absent or minimal in our samples, a substantial minority (28%) reported moderate to high levels of distress. High fascination with visual snow was reported by 9%. In Studies 2 and 3, visual snow was measured by means of verbal descriptions without graphic simulation ("visual field full of tiny dots of light" and "world seen with many dots of light", respectively). The results were similar to those in Study 1, but seeing visual snow 80%-100% of the time was less frequent (6.5% in Study 2 and 3.6% in Study 3). Visual snow has been insufficiently investigated. More research is needed to uncover underlying neurophysiological mechanisms and psychological and behavioral correlates.

## Introduction

Visual snow refers to a visual phenomenon characterized by tiny flickering dots occupying the entire visual field. The dots are often described as bright on dark backgrounds and dark on light backgrounds. They can also be described as brighter than the background or colorful [1–3]. Visual snow has attracted much attention in recent years [1–16], often with the implication

**Data Availability Statement:** All relevant data for this study are publicly available from the OSF repository (https://osf.io/sy5k8/).

**Funding:** The authors received no specific funding for this work.

**Competing interests:** The authors have declared that no competing interests exist.

that it is a pathological and rare condition. The term "visual snow syndrome" was proposed when it occurs with other visual phenomena, like palinopsia (after-images or visual trailing), photophobia (oversensitivity to light), nyctalopia (night-blindness), and entoptic phenomena (visual phenomena arising with sources in the eye [1–3]. Visual snow appears to be unrelated to ophthalmologic problems, but co-occurs frequently with migraine and tinnitus [2–7,12].

Most research on visual snow has been conducted with clinical samples. A subgroup of persons experiencing the visual snow phenomenon may be worried about it and seek clinical assessment of potentially serious causes. Only those patients then typically enter the clinical studies. Many people experiencing visual snow may not be worried by this experience, as indicated by numerous anecdotal reports on the internet. Even some people in clinical studies are not worried about it [17]. One study examined the prevalence and correlates in the general population. In a representative British sample, 3.7% of the respondents reported seeing visual snow (CI: 2.7–5.2), and 2.2% fulfilled the criteria for the visual snow syndrome (CI: 1.4–3.3) [18]. Respondents in this nonclinical representative sample who met the criteria for the visual snow syndrome were more likely to report severe headaches, tinnitus, and mood symptoms [18] than people without visual snow syndrome. This finding is congruent with the studies in clinical samples. The obtained prevalence of visual snow may be underestimated, as the study asked whether respondents experience visual snow "always or almost always". Visual snow may be a transient experience or even a natural phenomenon which many people sometimes perceive if attention is focused on it [19]. A higher prevalence of visual snow was suggested by a study with 117 participants with history of hallucinogen consumption divided in three groups. The prevalence of those with frequent or constant visual snow varied roughly between 16% and 21% before use of hallucinogens, precluding the possibility of visual snow being related to Hallucinogen Persistent Perception Disorder (HPPD) by that time [20].

Herein, we present the results of three studies examining the prevalence and correlates of visual snow in the general Portuguese population. We also wanted to find out to what extent visual snow causes not only distress, but also fascination, because many people are fascinated by particular peculiar visual phenomena.

## Attentional absorption and visual snow

An untested possibility is that people experiencing visual snow may be more engrossed in perceptual and imaginary experiences, because visual snow seems to result from hyperexcitability of the visual cortex [4–12]. If that is the case, such hyperexcitability could intensify visual perceptions and mental imagery, thereby facilitating attentional absorption in percepts and mental images. Studies show that vividness of mental imagery is related to activity of primary [21,22] and association visual cortex [22,23], suggesting that if visual cortex becomes hyperactive, imaginary experiences are likely to become more vivid, and as such more likely to grab attention, at least if attention was previously directed inwards. In accordance, pharmacologically-induced visual cortex hyperactivation through LSD correlates with increases in vividness of mental imagery, but not with other psychoactive effects [24]. Additionally, more responsiveness to the pattern glare task (an index of visual cortex hyperexcitability) was associated with history of out-of-body experiences, which can be seen as an extreme case of visual imagery vividness [25,26]. Pattern-glare task responsiveness was unrelated to history of sensed presence experiences, which are not characterized by vivid visual experiences, suggesting that the visual cortex hyperexcitability does not promote altered states of consciousness in general, but rather those characterized by strong visual imagery vividness [26].

A core characteristic of engrossment in mental imagery is imagination vividness [27–29], and there are reasons to think that visual snow may be associated with imagination vividness.

Visual snow is associated with hyperexcitability of the primary and association cortex [4–12], whose activity increases as visual imagery gets more vivid [21–24]. More specifically, visual snow has also been associated with greater activity [5–10] and greater gray matter volume of the lingual gyrus [8,15], an area of the associative visual cortex that tends to become more active when more vivid mental images are generated or when more vivid memories are recalled [30–33]. Also, the lingual gyrus was found to be larger in people with greater ability to imagine [34], and more active during tasks of internally directed attention [35]. Such increased activity of the visual cortex is thought to facilitate the awareness of visual snow, which might result from the conscious perceptions of a particular form of sensory information that normally remains subthreshold [5]. There is less evidence suggesting a link between visual cortex hyperactivity and intensified perception of external events, but engrossment in sensory perceptions is often accompanied by self-generated imagination [28,29]. In addition, a small study reported that people with visual snow have intensified sensory experiences in the olfactory and tactile modalities [36].

The individual differences in the propensity to be attentionally engrossed in sensory and imaginary experiences are reflected in the personality trait "absorption" [27–29,37–40]. This trait refers to the tendency to experience absorbed attentional states, i.e., intense states of focused, non-reflective, attention to perceptual or imaginary experiences [27–29,37–40]. Tellegen and Atkinson reported three overlapping aspects of absorbed states [27]: 1) a heightened sense of reality of the attentional object that to some extent excludes metacognitions (thinking about the attentional object), 2) imperviousness to normally distracting events as individuals become less aware of the surrounding space, and 3) an altered sense of reality. The experience of one part of reality is amplified, while the presence of other aspects, not only the environment, but also thoughts and characteristics of the self, retreat in consciousness. In retrospect, the experience might be remembered as fascinatingly strange or intriguing, or even as a discontinuity of the usual awareness of self and the world. A great sense of proximity to the attentional object is felt [27]. The type of focused attention that characterizes absorption is not the result of sustaining attention actively and with conscious effort; instead, absorbed states "have the quality of effortlessness, as if they happened by themselves, and in that sense, of involuntariness" (p. 222) [28]. Fascination is another term for the experience of absorption [27].

Higher propensity to experience absorbed states has been associated with openness to experience [38,39,41–43], appreciation of arts [43], rich and vivid imaginary [39,44,45], daydreaming as means to problem solving [46], greater hypnotic susceptibility [27,29,37,42,44,46], higher sexual desire [38,41], and mystical experiences [29,47]. People with the propensity for absorption are more prone to unusual perceptual experiences [39]. Also, the scales commonly used to measure trait absorption have an item that taps into whether people experience afterimages [27,37], which is a criterion for the proposed visual snow syndrome [1–3]. Hence, the experience of visual snow may well be related to the personality trait of absorption, but the possibility of such relationship was never examined.

To summarize, in the present study, we explored how frequently visual snow occurs in an unselected nonclinical group, and how frequently visual snow causes distress and fascination in this group. We also explored correlations between the frequency of seeing visual snow and tinnitus, and trait absorption.

## Method

### Procedure

Three studies on these research questions were performed in Portugal with online surveys advertised on social media between 2017 and 2021. These were part of a series of studies on

**Table 1. Measures used in Studies 1, 2, and 3.**

| | Study 1 | Study 2 | Study 3 |
|---|---|---|---|
| Visual snow (seen with open eyes) assessed with the aid of animated graphic simulation | X | | |
| Visual snow (seen with closed eyes) assessed with the aid of animated graphic simulation | X | | |
| Visual snow (seen with open eyes) described as a visual field full of tiny dots of light | | X | |
| Visual snow (seen with closed eyes) described as a visual field full of tiny dots of light | | X | |
| Visual snow (seen with open eyes) described as the world seen with many dots of light | | | X |
| Absorption | X | X | X |
| Blue field entoptic phenomenon assessed with the aid of animated graphic simulation | X | | |
| Floaters assessed with the aid of a picture | X | | |
| Distress caused by visual snow | X | X | |
| Fascination with visual snow | X | X | |
| Tinnitus | X | X | |
| Distress caused by tinnitus | X | X | |
| Fascination with tinnitus | X | X | |
| Situations triggering visual snow | X | X | |
| Life events associated with the first-time appearance of visual snow | X | X | |
| Analogies of visual snow | X | X | |
| Time of visual snow first appearance | X | X | |
| Intensification of visual snow over time | X | X | |
| Computer-screen time | X | X | |
| Ophthalmological problems | X | X | |
| Low blood pressure | X | | |
| Blood-pressure drops | X | | |
| Lifetime use of psychoactive substances | X | | |

correlates of attentional absorption. All studies received ethical approval from the local Ethics Committee. We will refer to the three studies as Study 1, Study 2, and Study 3. As described below, measures of visual snow differed across the studies because we were interested in comparing the prevalence of visual snow when visual snow is measured with the aid of an animated graphic simulation or by mere verbal descriptions. Table 1 illustrates the variables assessed in the three studies. Measures are described in the next subsection.

## Measures

**Measures of Study 1.** We showed animated simulations of visual snow and of a blue field entoptic phenomenon for 10 seconds. The simulations were taken from VisionSimulations. com. Participants were instructed to pay attention to the dots of light and to replay the video as often as they wanted. They were then asked how frequently they see similar dots of light occupying their entire visual field with their eyes open in everyday life and instructed to rate the frequency on a visual analogue scale by positioning a cursor on a line anchored by 0% (Never) and 100% (All the time). The line had 101 possible positions and responses were rounded to fit percentages from 0% to 100% in 11 steps (0%, 10%, 20%, . . ., 100%). The simulation of visual snow can be found at https://visionsimulations.com/visual-snow.htm?background=restaurant.jpg. The simulation of the blue field entoptic phenomenon can be found at https://visionsimulations.com/blue-field-phenomena.htm.

Participants were also asked to report how frequently they see the dots of light representing visual snow when their eyes are closed using the same visual analogue scale.

Participants were shown a graphic representation of floaters (also known as *muscae volitantes*) and instructed to look at the appearing floater forms. They were then asked to rate how frequently they see similar forms in daily life using the same visual analogue scale.

The degrees of distress and fascination caused by visual snow were measured on a 7-point Likert scale ranging from 1 (absolutely nothing) to 7 (extremely).

Two open-ended questions were included asking about situations that commonly trigger the appearance of visual snow and if there was a life event that could be associated with the first-time appearance of visual snow. If such triggering situations and life events existed, respondents were asked to describe them in their own words.

Participants were requested to tick options indicating analogies for their experience of visual snow (they could tick more than one option). Options were: a) static of an analog television, b) snowfall, c) uncertain, d) other analogies. If they chose "other analogies", they were asked to describe them.

We inquired since when visual snow has been seen. Options of answers were: a) since childhood, b) since some years ago, and c) since very recently. Answers b and c were collapsed into a category "since after childhood".

We asked about the intensification of visual snow with time. Options of answers were: a) the intensity has been increasing with time, b) the intensity has been decreasing with time, and c) the intensity has remained the same.

The computer-screen time was assessed by asking participants to estimate the number of hours per day they spend looking at a computer screen.

The frequency of tinnitus was assessed by asking participants about the amount of time they hear a ringing, buzzing, hissing, or other sounds that do not come from the external world. Answers were given on the same scales used to assess visual snow frequency, i. e., the frequency of experienced tinnitus was reported on a scale from 0% (Never) to 100% (All the time) with intervals of 10.

The levels of distress and fascination caused by tinnitus were measured on a 7-point Likert scale from 1 (absolutely nothing) to 7 (extremely).

Participants were asked if they have ophthalmological problems, low blood pressure, and episodes of blood pressure drops.

Lifetime use of psychoactive substances was assessed for the following classes of substances: cannabis/synthetic cannabinoids, MDMA, psychedelics, cocaine/stimulants.

The propensity for attentional absorption was measured with the Modified Tellegen Absorption Scale (MODTAS) [37], which is composed of 34 self-descriptive items to be rated on a 5-point scale from 1 (I totally disagree) to 5 (I totally agree), which are assigned to the following five subscales [37]:

1. Imaginative involvement: propensity to be attentionally absorbed in imagination. Examples of items: "If I wish I can imagine (or daydream) some things so vividly that they hold my attention as a good movie or story does."; "I can sometimes recollect certain past experiences in my life with such clarity and vividness that is like living them again or almost so".

2. Esthetic involvement in nature: propensity to be attentionally absorbed in external events. Examples of items: "It is sometimes possible for me to be completely immersed in nature or in art and feel as if my whole state of consciousness has somehow been temporarily altered"; "When I listen to organ or other powerful music, I sometimes feel as if I am being lifted into the air".

3. Altered states of consciousness: tendency to experience psychological states that deviate markedly from the usual waking consciousness. Examples of items: "I think I really know

what some people mean when they talk about mystical experiences"; "Sometimes I experience things as if they were doubly real".

4. Synesthesias: proneness to experience associative synesthesias [48], i. e., when sensations in one particular sensory modality elicit mental associations in a different sensory modality, normally visual. Examples of items: "I find that different odors have different colors"; "Some music reminds me of pictures or changing color patterns".

5. Extrasensory perception (ESP): tendency to be impressed by events that are intriguing because they apparently defy known physical laws. Examples of items: "I can often somehow sense the presence of another person before I actually see her/him"; "I often know what someone is going to say before he or she says it".

**Measures of Study 2.** Visual snow was assessed by asking participants to report the amount of time their visual field is full of tiny dots of light when their eyes are open (here we did not use visual simulations). Answers were given on an 11-point scale with options 0% (Never), 10%, 20%, . . . 100% (All the time). We repeated the same question with the same scale to assess the frequency of visual snow with the eyes closed. We described visual snow as "dots of light" because that description is what is actually seen in most graphic simulations. As we will show in the Results section, not all people associate their experience of visual snow with static of analog television.

Measures similar to those in Study 1 were used to assess absorption, tinnitus frequency, tinnitus distress, tinnitus fascination, situations triggering visual snow, life events associated with the first appearance of visual snow, analogies of visual snow, intensification of visual snow over time, first-time appearance of visual snow, ophthalmological problems, and computer-screen time.

**Measures of Study 3.** Visual snow frequency was assessed as the amount of time the world was seen with many dots of light. Answers were given on an 11-point scale with options 0% (Never), 10%, 20%, . . . 100% (All the time).

Absorption was measured like in Studies 1 and 2.

## Results and discussion

### Prevalence of visual snow

Sample characteristics and the prevalence of visual snow across studies are reported in Table 2. Prevalence of visual snow is also graphically depicted in Fig 1. Information regarding other variables of interest can be found in S1, S2 and S3 Tables.

The prevalence of those who never see visual snow was remarkably similar regardless of whether the assessment was performed with the aid of animated graphic simulations or participants were simply asked about the frequency of seeing "dots of light" (roughly 55%-56%). This means that roughly 45% of the respondents reported some visual experience of snow with varying degrees of frequency. In Study 1, 14% [95% CI: 11% - 17%] of the respondents reported seeing visual snow permanently (100% of the time). The percentage of those who always or almost always see visual dots (between 80% and 100% of the time) was about 20% (95% CI: 16% - 23%). Study 1 used animated graphic simulations of visual snow. When participants were asked about the frequency of seeing visual snow without the aid of graphic simulations, the proportion of those reporting seeing "dots of light" always or almost always (between 80% and 100% of the time) was lower: 6.5% (95% CI: 4% - 10%) in Study 2 and 3.6% (95% CI: 2% - 6%) in Study 3. Visual snow may be a rather common phenomenon, but some

**Table 2. Sample characteristics and frequency of visual snow.**

| | Study 1[a] | Study 2[b] | Study 3[c] |
|---|---|---|---|
| *N* | 564 | 310 | 368 |
| Women | 332 | 240 | 250 |
| Men | 232 | 70 | 118 |
| *Age* | | | |
| Mean (SD) | 32.43 (16.68) | 30.77 (11.57) | 27.18 (8.53) |
| Range | 18–75 | 18–67 | 18–75 |
| Interquartile range | 18 | 17 | 6 |
| *Measure of visual snow* | % of time the visual field looks like an animated simulation of visual snow | % of time the visual field is full of tiny dots of light | % of time the world is seen with many dots of light |
| *Estimated percentage of time seeing visual snow* | 55.9 | 54.5 | 56.0 |
| 0% | 7.6 | 9.0 | 15.2 |
| 10% | 4.3 | 6.8 | 6.3 |
| 20% | 3.9 | 5.2 | 6.5 |
| 30% | 2.0 | 2.3 | 3.5 |
| 40% | 3.7 | 6.5 | 3.8 |
| 50% | 1.6 | 5.2 | 2.7 |
| 60% | 1.4 | 4.2 | 2.4 |
| 70% | 3.2 | 2.3 | 1.4 |
| 80% | 2.1 | 1.6 | .8 |
| 90% | 14.4 | 2.6 | 1.4 |
| 100% | | | |
| *Percentage of time seeing visual snow with closed eyes* | 30.2 | 29.7 | |
| 0% | 13.4 | 11.7 | |
| 10% | 12.0 | 5.9 | |
| 20% | 8.7 | 9.7 | |
| 30% | 6.8 | 7.6 | |
| 40% | 9.1 | 10.3 | |
| 50% | 5.2 | 5.9 | |
| 60% | 5.2 | 4.8 | |
| 70% | 3.1 | 5.2 | |
| 80% | 3.1 | 3.4 | |
| 90% | 3.1 | 5.9 | |
| 100% | | | |
| Visual snow present since childhood (subsample of those with visual snow) | 47.4 | 50.5 | |
| *Progression of visual snow (subsample of those with visual snow)* | | | |
| It has been increasing with time | 7.7 | 15.3 | |
| It has been decreasing with time | 12.2 | 6.1 | |
| It has remained constant | 75.1 | 78.6 | |

[a] collected in 2020–2021

[b] collected in 2018–2019

[c] collected in 2017.

[1] In the group of those not seeing visual snow, a small proportion (N = 13; 2.4% of the total sample in Study 1) reported seeing it very occasionally.

people only notice it when instructed to pay attention to it, and the graphic simulation may have been more effective in calling attention to the fact that visual snow is "permanently or usually there". A similar pattern can be observed with entoptic phenomena, which may only become visible after attention has been called to them. The use of graphic simulations is likely

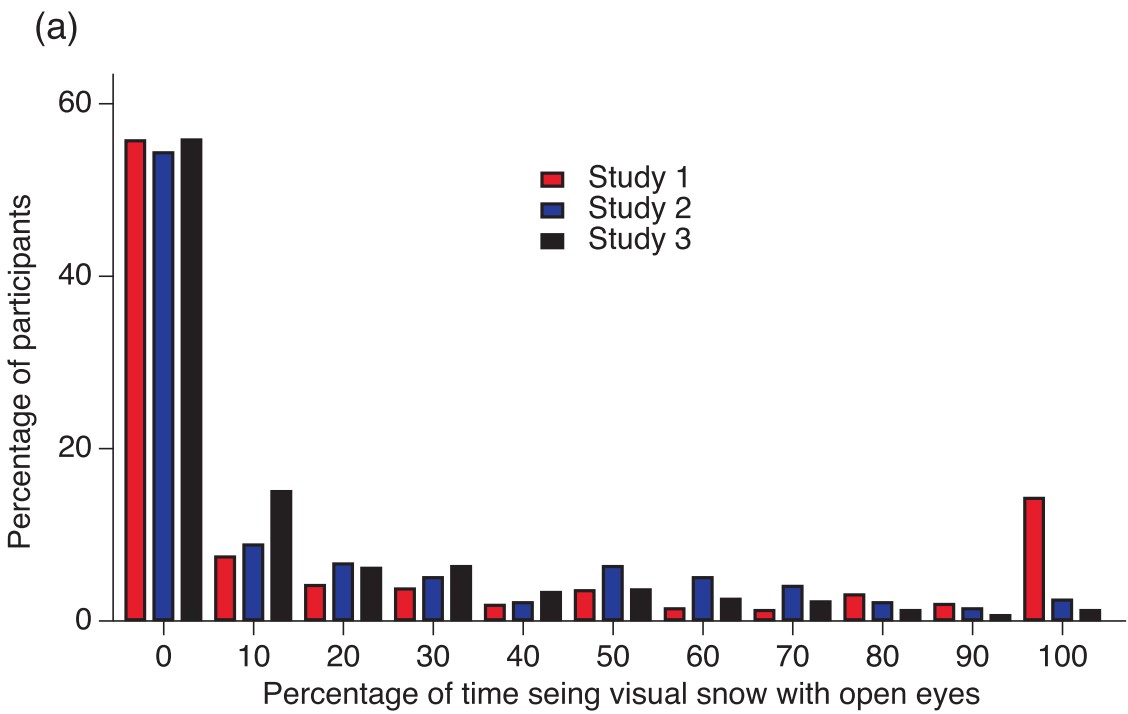

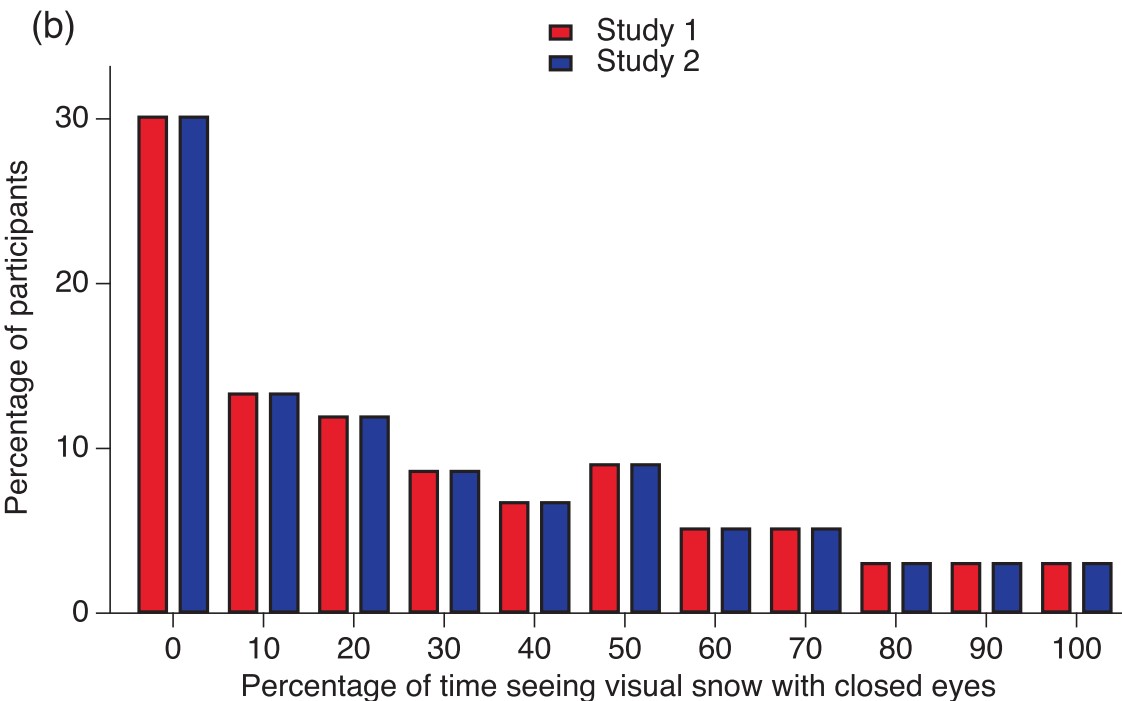

**Fig 1. Frequency of visual snow.** (a) Percentage of participants seeing visual snow with open eyes for a certain amount of time. (b) Percentage of participants seeing visual snow with open eyes for a certain amount of time. Red bars: Data from Study 1. Blue bars: Data from Study 2. Black bars: Data from Study 3.

a more reliable method because it does not depend on descriptions of particular analogies. Therefore, it is possible that the results of Studies 2 and 3 are underestimations.

In a study with a representative British sample, the prevalence of visual snow occurring always or almost always was estimated at 3.7% (CI: 2.7–5.2) [18], which is similar to the results obtained in Studies 2 and 3. The British study did not use a graphic representation, but simply asked, "Do you have visual symptoms that can be described like this: 'tiny dots, continuously occurring in the entire visual field? The dots are usually black/grey on white background or grey/white on a black background, but they can also be transparent, white flashing or colored" [18]. As noted above, when graphic simulations are not employed, many respondents might not associate their experience with the verbal descriptions offered. As shown in Table 3, only 67% of the respondents in Study 1 who experience visual snow reported that it resembled the static of analog television, which is the analogy most commonly given. One third of the sample did not make the association with static. One quarter was uncertain about what analogy to use (see Table 3).

Notably, the prevalence of frequent or persistent visual snow we obtained in Study 1 is similar to that obtained in a study of users of hallucinogens before they started using this type of drugs [20], which rules out that hallucinogens were part of the etiology.

The results of the present study must be interpreted with caution. Our samples were not representative of the Portuguese population, although in comparison with other studies, the total number of respondents was large ($N = 1,242$). Future studies with more representative samples and stratified according to gender, education, and age are needed. The results still suggest a higher prevalence of visual snow in the general population than is often assumed and also indicate that visual snow is not an all-or-nothing phenomenon, i.e., it is not permanently present in the visual field of those who experience it. Visual snow appears to be more

**Table 3. Analogies of visual snow (Study 1; N = 209).**

| | |
|---|---|
| Static of an analog television | 67%[1] |
| Snowfall | 7.7%[1] |
| Uncertain | 24.9%[1] |
| Other analogies | 6.2%[1] |
| *Description of the other analogies* | |
| Grain in photos (not static from analog television) | .2%[2] |
| Dark moving dots (in addition to luminous static) | .2%[2] |
| Non-luminous static with moving spots (in addition to luminous static) | .2%[2] |
| Out-of-focus image | .2%[2] |
| Tiny colorful dots | .2%[2] |
| Dust particles | .2%[2] |
| Rounded clouds | .2%[2] |
| As if seeing the wind | .2%[2] |
| Fireflies | .2%[2] |
| Falling purpurins | .2%[2] |
| Sparks of the reflection of sunlight on water | .2%[2] |
| As if cells falling from heaven | .2%[2] |
| Angels | .2%[2,3] |

[1] Sum of percentages is more than 100% because some respondents ticked more than one option.

[2] Percentages of .2 refer to one single participant for each description.

[3] This respondent reported that visual snow appeared during a process of religious transformation. This is more accurately described as a subjective interpretation than an analogy.

frequently seen with closed eyes [36]. In Studies 1 and 2, around 70% reported seeing visual snow at least occasionally with closed eyes (see Table 2 and Fig 1).

## Visual snow and tinnitus

In clinical groups, tinnitus is a consistent correlate of visual snow, and the two conditions may share common etiologies. This finding was replicated in the nonclinical groups in the present investigation. Table 4 shows a significant positive correlation between visual snow frequency and tinnitus frequency in Studies 1 and 2. There exists a wide variability in the frequency of both tinnitus and visual snow (see Fig 2), but tinnitus was experienced by a larger proportion of respondents than visual snow. Tinnitus was reported to be experienced at least 10% of the time by 58% of respondents in Study 1 and 48% in Study 2 (see Fig 2). This is larger than the reported prevalence of tinnitus that ranges between 5% and 43% [49]. However, there are no standardized measures for tinnitus, which makes prevalence estimates uncertain. Research on tinnitus prevalence has resorted to different types of questions across studies [49]. Although we cannot rule out that our measure of tinnitus overestimated its prevalence, it is not equally possible to rule out that other measures underestimated it, especially when it occurs briefly and/or infrequently. For example, sometimes questions focus on tinnitus ("noises in your

**Table 4. Correlates of visual snow (Pearson correlation coefficients).**

|  | Study 1 | Study 2 | Study 3 |
|---|---|---|---|
| Sex (women = 1, men = 2) | .08 | -.05 | .14* |
| Age | .01 | .01 | .09 |
| Tinnitus frequency | .23*** | .37*** |  |
| Blue-field entoptic phenomenon | .39*** |  |  |
| Floaters | .21*** |  |  |
| Visual snow since childhood[1] | .02 | -.06 |  |
| Distress caused by visual snow | .37*** | .18* |  |
| Fascination with visual snow | -.03 | .32*** |  |
| Absorption (MODTAS total score[2]) | .17*** | .27*** | .34*** |
| MODTAS Imaginative involvement[2] | .13** | .24** | .28*** |
| MODTAS Esthetic involvement in nature[2] | .11* | .21** | .26*** |
| MODTAS Altered states of consciousness[2] | .24*** | .33*** | .35*** |
| MODTAS Synesthesias[2] | .13** | .21** | .28*** |
| MODTAS Extrasensory Perception[2] | .15** | .24** | .35*** |
| Ophthalmological problems | .02 | .05 |  |
| Computer-screen time | -.04 | -.03 |  |
| Low blood pressure | .01 |  |  |
| Blood-pressure drops | .07 |  |  |
| Lifetime use of cannabis | .02 |  |  |
| Lifetime use of psychedelics | .02 |  |  |
| Lifetime use of MDMA | -.001 |  |  |
| Lifetime use of cocaine/stimulants | .03 |  |  |
| Lifetime use of any the psychoactive substances listed above | .02 |  |  |

[1] 0 = visual snow present since after childhood: 1 = visual snow present since childhood.

[2] MODTAS = Modified Tellegen Absorption Scale.

* $p < .05$

** $p < .01$

*** $p < .001$.

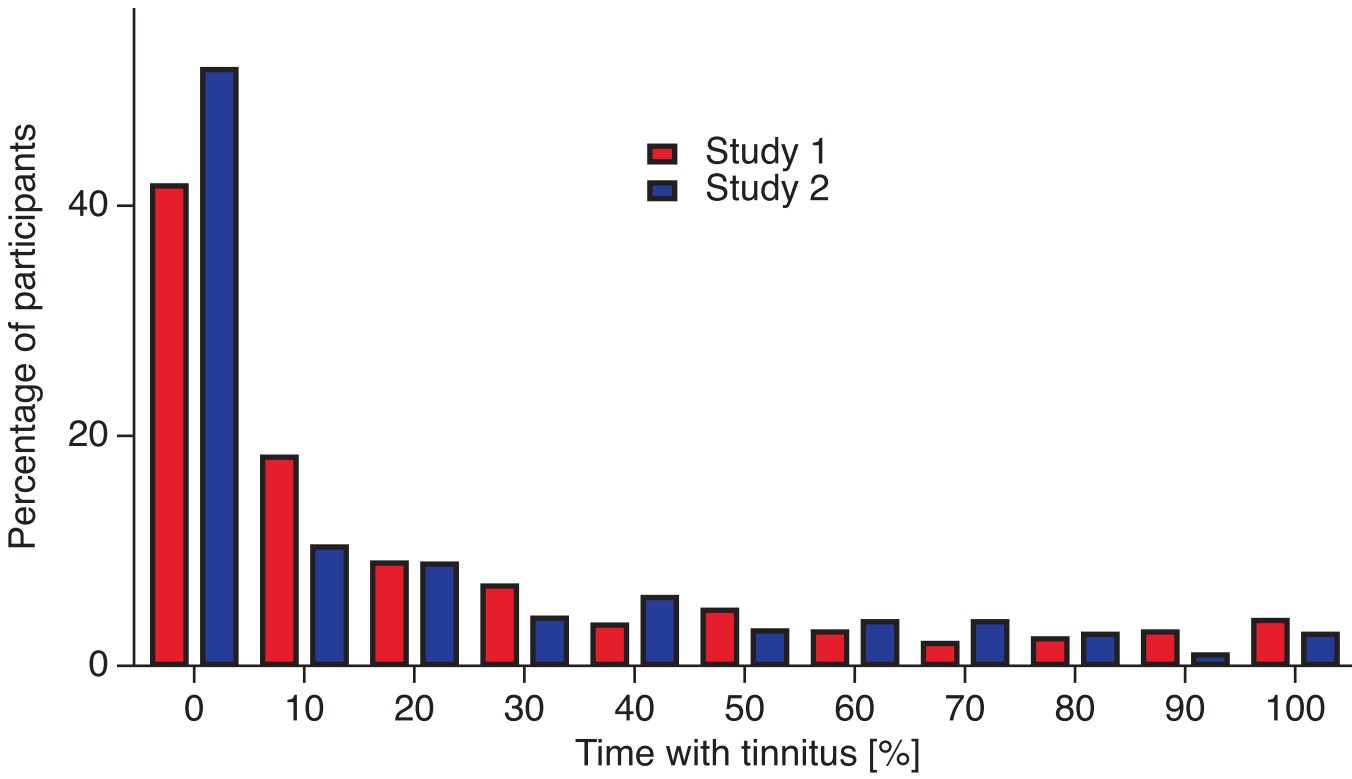

**Fig 2. Frequency of tinnitus.** Percentage of participants experiencing tinnitus for a certain amount of time. Red bars: Data from Study 1. Blue bars: Data from Study 2.

head or ears") for more than five minutes, which may not always happen. In contrast, our measure assesses a large spectrum of tinnitus frequencies without making reference to tinnitus duration and including very transient and infrequent experiences. In fact, when we do not count the participants reporting frequencies of 10% (see Fig 2 and S2 Table), the prevalence is inside the reported range [49]. The proportion of respondents experiencing tinnitus more than 10% of time was 40% in Study 1 and 37% in Study 2 (see Fig 2 and S2 Table), which is comparable to that of the literature [49].

Other commonly used measures ask if within the past year, respondents did ever hear a sound ("buzzing, hissing, ringing, humming, roaring, machinery noise") originating in the ear [49]. It is possible that when tinnitus is infrequent, many people do not report it, because they do not want to identify as someone who hears noises that do not have an external origin. Because our measure allows the option of responding affirmatively to the existence of infrequent tinnitus, it may avoid such identification. Only future research comparing different types of measures could clarify these issues.

### Reactions to visual snow and tinnitus

The emotional responses (distress and fascination) provoked by visual snow were quite similar regardless of whether visual snow had been assessed with the aid of animated graphic simulations or not. In Studies 1 and 2, the levels of distress resulting from visual snow varied considerably (see Fig 3), but most respondents (between 63% and 69%) do not feel distressed, and if they do, the distress is minimal. Between 30% and 32% of those seeing visual snow reported absolutely no distress (scoring 1 point on the 7-point distress scale). Between 37% and 38%

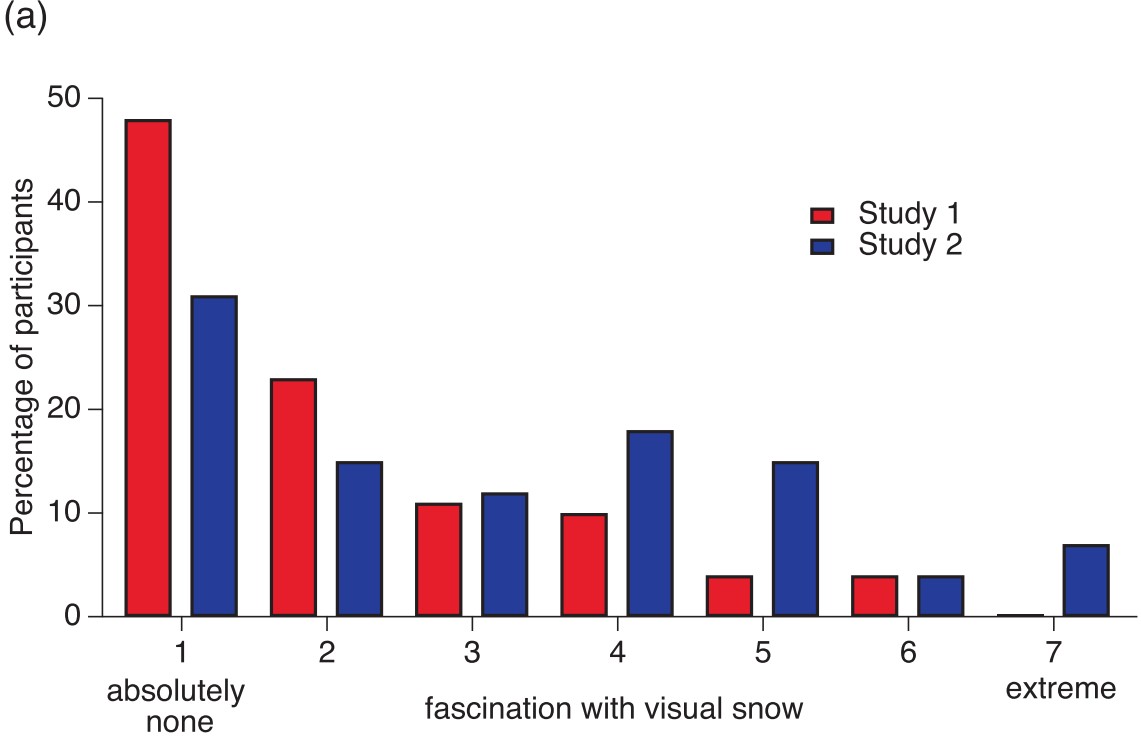

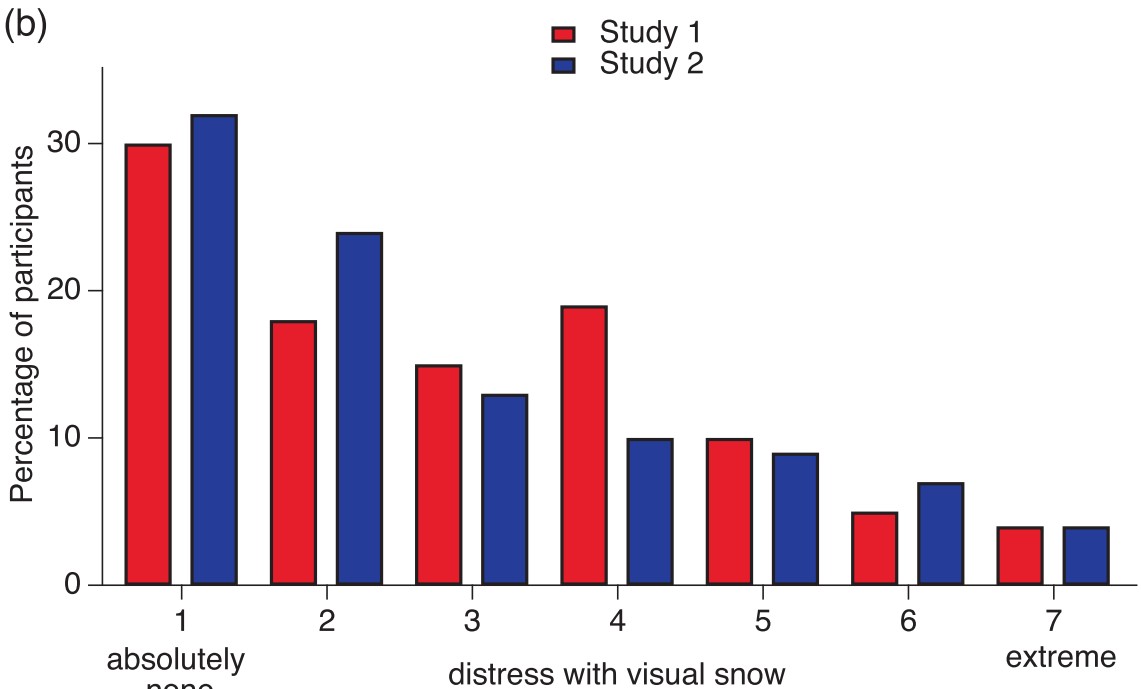

**Fig 3. Distress and fascination with visual snow.** (a) Percentage of participants with different degrees of distress with visual snow. (b) Percentage of participants with different degrees of fascination with visual snow. Red bars: Data from Study 1. Blue bars: Data from Study 2.

reported minimal or little distress (scoring 2 and 3 points). Between 10% and 19% reported moderate distress (scoring 4 points). Between 18% and 20% complained of higher levels of distress (scoring between 5 and 7 points). Although many people are not distressed by their visual snow, there appears to be a substantial proportion that could possibly benefit from clinical attention. As with tinnitus, pharmacologic interventions for visual snow are largely ineffective. Possible strategies to help people who are distressed by visual snow are cognitive-behavioral therapies like those used for tinnitus distress. The objective would not be to eliminate visual snow, but to eliminate the associated distress [50].

The distribution of distress caused by visual snow is similar to that of distress with tinnitus (see Fig 4). Most respondents with tinnitus were not distressed at all, and if they were, distress was minor. Between 22% and 30% reported higher levels of distress with tinnitus (scoring between 5 and 7 points on the 7-point distress scale). It is well known that many people are not distressed by tinnitus [49–51]. For comparison with reported prevalence of bothersome tinnitus, we calculated the proportion of respondents reporting moderate to severe distress with tinnitus from the total sample (including those without tinnitus). We obtained a proportion of 17% in Study 1 and 20% in Study 2. The prevalence of bothersome tinnitus reported in the literature ranges from 3% to 30% [49].

Because sometimes unusual visual phenomena cause fascination, we investigated whether visual snow also evokes this positive reaction. Some participants reported higher degrees of fascination with visual snow; 9% in Study 1 and 24% in Study 2 scored between 5 and 7 on the 7-point fascination scale (see Fig 3). Fewer people reported fascination with tinnitus (see Fig 4).

As shown in Table 4, visual snow frequency correlated with greater associated distress in Studies 1 and 2, but more strongly so in Study 1. Visual snow frequency correlated with fascination moderately in Study 2, and was uncorrelated in Study 1. It seems that greater frequency of visual snow may contribute to distress in some people, but many others do not feel uncomfortable with the persistent experience. In a multiple regression done with the Study 2 sample, we found that distress ($\beta = .24$, $p = .004$) and fascination ($\beta = .36$, $p < .001$) were directly and independently associated with visual snow frequency. More research is needed to clarify how the frequency of visual snow correlates with psychological reactions to it. We cannot exclude that the discrepancy in Study 1 and 2 is due to some bias in sample selection or to different aspects of visual snow being measured, as the studies used different forms of assessment. To understand why some people are distressed, and others are not is a crucial one. Comparing groups of people with and without distress might bring clarification to this important question.

## Visual snow and entoptic phenomena

Study 1 demonstrated that visual snow frequency correlated directly with the frequency of two entoptic phenomena: floaters and blue field entoptic phenomenon (see Table 4). This coincides with research in clinical samples in which visual snow and entoptic phenomena often occur together. Schankin and colleagues observed that blue field entoptic phenomena occurred in 79% of a group of patients with visual snow; floaters occurred in 81% [2]. In fact, entoptic phenomena are among the additional visual phenomena required to diagnose the visual snow syndrome [1–3]. Even in nonclinical groups, more people report visual snow with additional visual phenomena than without [3,18].

Given the lack of standardized measures to assess entoptic phenomena, we must be cautious regarding the obtained prevalence (see S3 Table). However, we believe that our measure has advantages over others, as we used graphical simulations and resorted to a scale that

(a)

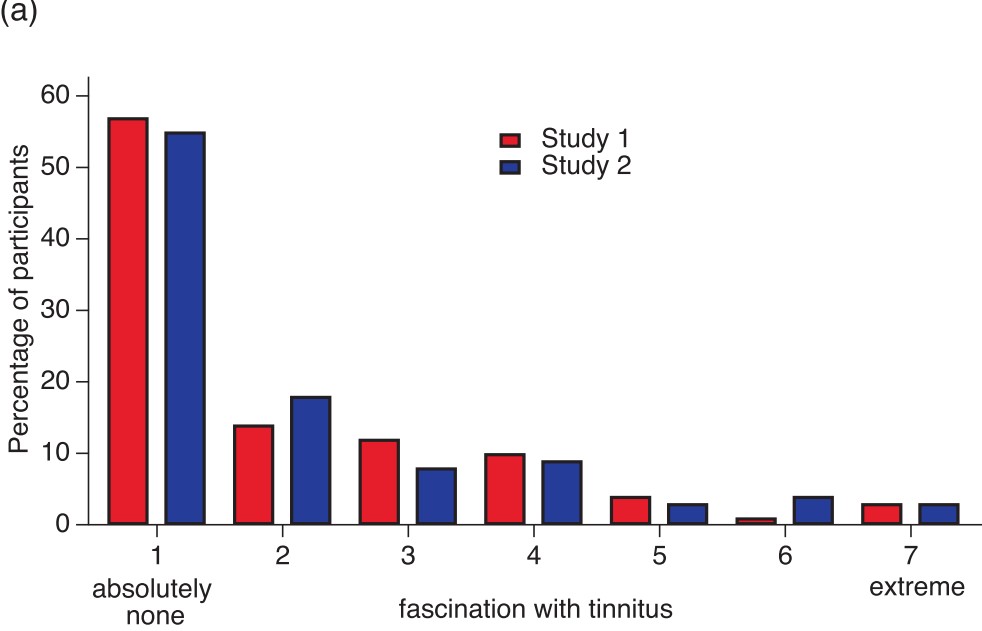

(b)

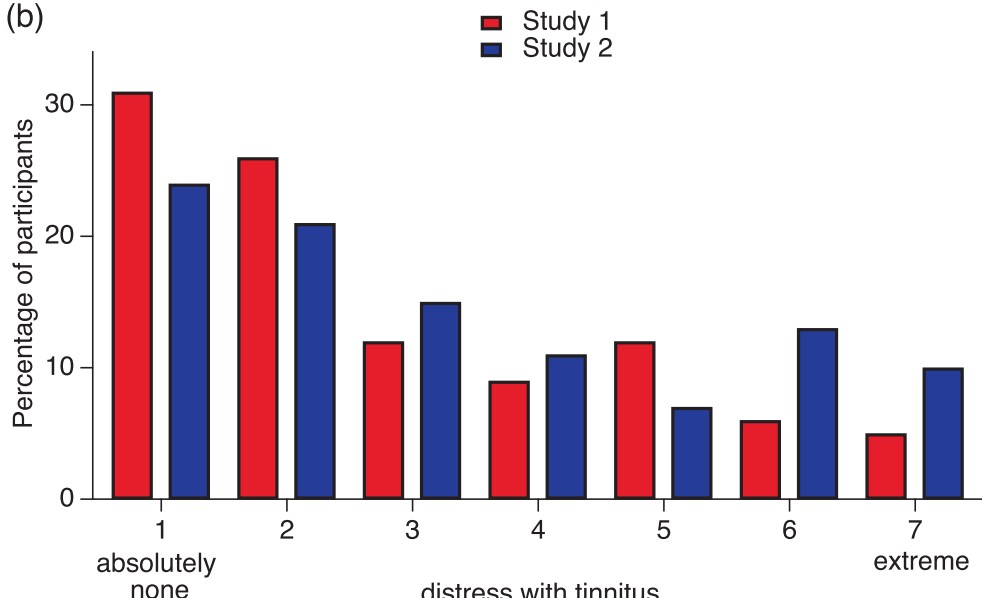

**Fig 4. Distress and fascination with tinnitus.** (a) Percentage of participants with different degrees of distress with tinnitus. (b) Percentage of participants with different degrees of fascination with tinnitus. Red bars: Data from Study 1. Blue bars: Data from Study 2.

encompasses a large spectrum of frequency of the experiences (including only occasional experiences). These are likely an improvement over measures that only rely on verbal descriptions and less nuanced scales of frequency of occurrence of entoptic phenomena. Only future research using different measures, including in-person interviews, might clarify the question of the real prevalence.

## Triggers of visual snow

Because many people who see visual snow do not see it all the time, it is important to ascertain if there are situations that trigger short-term appearances of visual snow. Only some respondents with visual snow reported such triggers (31% in Study 1 and 26% in Study 2 among those seeing visual snow). As shown in Tables 5 and 6, we detected eight types of triggers: light-related, attention-related, tiredness-related, blood pressure-related, mood-related, eye-related, migraine-related, and pain-related. For those reporting light-related triggers, visual snow appears when looking at intense lights, when changing from dark to bright environments or when being in dark surroundings. Attention-related triggers refer to situations in which visual snow appears as a result of highly focused attention on something, but "vague thoughts" or "looking at the void" can also trigger visual snow, which indicates rather dispersed attention. Attention-related and light-related triggers can overlap, as visual snow can appear when focusing attention on lights. Visual snow can also appear when one is tired. Visual snow can become visible when drops in blood pressure are felt or as a consequence of movements that lower blood pressure. Mood-related triggers are more common with negative mood changes. Eye-related triggers are the result of a variety of physiological processes in the eyes, such as making pressure on the eyes or feeling "tired eyes". All these six types of triggers were equally observed in Studies 1 and 2, regardless of whether animated simulations of visual snow were presented or not (see Tables 5 and 6). Although migraine is associated with visual snow in clinical studies, only two participants in Study 1 associated migraine attacks with the appearance of visual snow. Four additional participants in Study 1 reported visual snow triggered by pain (headache in one case, back pain in one, and unspecified in the two others).

**Table 5. Triggers of visual snow (Study 1; N = 78): Number of participants reporting particular types of triggers are indicated in parentheses (some participants reported more than one type)[1].**

| Light-related[2] | Attention-related |
|---|---|
| Intense lights (8) | Focusing visual attention on something (6) |
| Sunlight (5) | Turning the attention to the visual snow (2)[3] |
| Focusing attention on lights[2] (4) | Changing attentional focus (1) |
| Light foci[2] (2) | Introspection (1) |
| Looking at tv or computer for a long time[2] (4) | Doing nothing (1) |
| Darkness/dim light (7) | |
| Passing from dark to bright environment (5) | **Tiredness-related** |
| Changes in luminosity (7) | Feeling tired (21) |
| **Mood-related** | **Migraine-related** |
| Anxiety (7) | Migraine attacks (2) |
| Stress (3) | |
| Strong emotions (1) | **Pain-related** |
| | When in pain (4) |
| **Blood pressure-related** | **Eye-related** |
| Blood-pressure drops (1) | Pressure on the eyes (2) |
| Raising quickly (4) | Tired eyes (3) |
| Sudden movements (2) | Not wearing glasses (1) |
| Feeling dizzy (1) | Unfocusing the eyes (1) |

[1] Other reported triggers were hot environments and when awakening.

[2] Some light-related triggers are also attention-related.

[3] One respondent reported that paying attention to visual snow is easier in nature.

**Table 6. Triggers of visual snow (Study 2; N = 35): Number of participants reporting particular types of triggers is indicated in parentheses (some participants reported more than one type).**

| Light-related | Attention-related[1] |
|---|---|
| Intense lights (9) | Focusing visual attention on something (4) |
| Sunlight (5) | Looking at the sky (1) |
| Focusing attention on lights[1] (4) | Vague thought (1) |
| Watching television in the dark[1] (1) | "Looking at the void" (1) |
| Looking at television or computer screens for a long time[1] (1) | Evoking memories/talking about something in the past (2) |
| Night darkness (1) | |
| Passing from dark to bright environment (1) | |
| **Mood-related** | **Tiredness-related** |
| Relaxation (1) | Feeling tired (1) |
| Stress (1) | |
| **Blood pressure-related** | **Eye-related** |
| Blood pressure drops (2) | Pressure on the eyes (1) |
| Raising the head quickly (4) | |
| Sudden movements (2) | |
| After sudden physical efforts (1) | |
| After physical efforts (1) | |

[1] Some attention-related triggers are also light-related.

Although reductions in blood pressure were a relatively common trigger, visual snow frequency was unrelated to self-reported low blood pressure and drops in blood pressure. Tiredness was a common trigger, especially in Study 1. Because fatigue has been associated with hypotension [52,53], future research is needed to ascertain the interrelationships between visual snow, fatigue and low blood pressure.

Intense light was a common trigger, and some participants specifically indicated that looking at television or computer screens elicited visual snow. We tested whether computer-screen time correlated with visual snow frequency, but found no supportive evidence. In Studies 1 and 2, computer-screen time was uncorrelated with visual snow frequency (see Table 4).

Since visual snow has various triggers, it is unclear if it has always the same underlying neural mechanisms, and if these are the same occurring in untriggered visual snow; for example, it is not possible to know if visual cortex hyperexcitability is always present. One way to get a clearer picture may be to apply measures of neural activity like EEG or fMRI and compare the phenomena and their neural correlates under different induction conditions, if this is possible in the lab. This may be combined with the development of a specific questionnaire about visual snow that contains a particular focus on the comparison of different inducing conditions.

We inquired about events that respondents associated with the first appearance of visual snow. Very few reported any such event, and the answers were very heterogeneous, as shown in Table 7. Some individuals may have reported non-causative, coincidental events. Only one respondent associated the use of substances (cannabis and synthetic cannabinoids) with the first-time appearance of visual snow. Table 4 indicates that lifetime use of a variety of psychoactive substances was unrelated to visual snow frequency. We also found visual snow to be unrelated to ophthalmological problems (see Table 5), which is consistent with the extant literature about visual snow. However, three participants associated the first appearance of visual snow with ophthalmological problems, which raises the possibility that some etiologies of visual snow might be related to eye disorders.

**Table 7. Events associated with the first appearance of visual snow: Numbers of participants reporting a particular event are indicated in parentheses.**

| |
|---|
| **Study 1 (N = 12)** |
| Ophthalmological problems (3) |
| Migraine (2) |
| Panic attacks (4) |
| "Waking in the direction of the sunset, as a child" (1) |
| Watching the sea (1) |
| Religious conversion to Christ (1)[1] |
| **Study 2 (N = 6)** |
| Use of cannabis and synthetic cannabinoids (1) |
| Intensive practice of sports (1) |
| Loss of family member (1) |
| Change to vegan diet (1) |
| During a hospital stay in childhood, when walking at night in the hospital corridors (1) |
| In childhood, dots of light appeared associated with fear of dark, and were imagined to be witches. They disappeared and reappeared later during university classes when attention was very focused on teachers to the point that it was like surrounding space did not exist (1) |

[1] This respondent associated visual snow with angels as an analogy (see Table 5).

The finding that very few participants could remember any event associated with the origin of visual snow corresponds with the study by Puledda and colleagues, in which 40% - 46% reported seeing visual snow since childhood (for as long as they could recall). The majority of those in whom it started after childhood could not recall any associated event [3]. The associated events, when recalled, varied [3]. Many participants in Study 1 (47%) and Study 2 (51%) reported that they have visual snow since childhood, as depicted in Table 2. These patterns were similar to those obtained by Puledda and colleagues [3]. Table 4 shows that visual snow frequency was uncorrelated with seeing visual snow since childhood.

Table 2 demonstrates that visual snow intensity usually remains stable. In a study by Schankin and colleagues, 41% of participants reported that their visual snow intensity had been stable from the start [2]. Among those with an initial phase of progressive intensification, 48% reported a later stabilization. Among those with a stepwise beginning, 100% reported that stabilization followed [2].

## Visual snow and absorption

Table 7 shows that the frequency of visual snow correlated positively with trait absorption, as well as with all the five dimensions: imaginary involvement, synesthesias, sensory involvement, extrasensory perception, and altered states of consciousness. This was found in all three studies, regardless of how visual snow was assessed. These correlations were largely explained by persons who never see visual snow having lower absorption in comparison with those who experience it at least 10% of the time. When the analyses were restricted to the subgroup seeing visual snow at least 10% of the time, the correlations between visual snow and absorption (and absorption dimensions) became nonsignificant or significant with a reduced effect size.

Point-biserial correlations were run between absorption and visual snow (dichotomously measured as no visual snow vs. seeing visual snow at least 10% of the time). With this approach, visual snow was always significantly associated with absorption (and absorption dimensions): Study 1: $.14 \leq r \leq .26$, all $p \leq .004$; Study 2: $.19 \leq r \leq .32$, all $p \leq .001$; Study 3: $.26 \leq r \leq .37$, all $p < .001$ (see S4 Table). In these point-biserial correlations with visual snow

dichotomously considered, altered states of consciousness continued to be the dimension of absorption most strongly correlated with visual snow. Thus, absorbed states do not seem to be associated with persistent visual snow, but rather with some susceptibility to experience it.

More frequent visual snow percepts were associated with a greater capacity to be engrossed in sensory and imaginary experiences that are perceived as fascinating. The correlation with the dimension "altered states of consciousness" was always the strongest in the three studies, raising the possibility that people with visual snow might be more susceptible to hypnosis [27,29,37,42,44,46], more open to enrapturing esthetic experiences [29], feel the effects of psychoactive substances more intensely [29], have more blissful sexual experiences [54], and be more prone to mystical-type experiences [29,47], which have recently become a topic of growing scientific interest [29,47,55–60]. These are important issues for future research, especially because some altered states of consciousness can be very enriching [54,58,60].

Across the three studies, the five dimensions of the MODTAS were strongly correlated with each other (.45 ≤ r ≤ .74, all p < .001). After collapsing the MODTAS data from the three studies, Cronbach's alpha for the total scale was .95. Acceptable results were obtained in an exploratory factor analysis using the principal components method with varimax rotation, carried with the 34 items of the MODTAS and requesting the extraction of one single factor. This factor explained 36.85% of variance (KMO = .97; Bartlett's Test of Sphericity: approximate $\chi^2$ = 6352.84, df = 561, p < .001). For 33 items, loadings ranged between .50 and .72. Item 2 had a loading of .40. If these analyses were done separately for the three studies, results were similar (see S5 Table). This indicates that the MODTAS items express a common latent variable: engrossment in imaginary and perceptual experiences with this kind of immersion in perceptual experiences being a rather passive and receptive experience, largely involuntary, like in states of fascination and wonder [27–29].

The MODTAS is a commonly used measure of absorption with results confirming its criterion validity [20,37,54,61–67], including in experimental [62,65,66], and longitudinal studies [63].

A possible explanation for the relationship between absorption and visual snow is visual cortex hyperexcitability that may facilitate visual snow and intensify perceptual and imaginary experiences to the extent that they absorb attention to a high degree. In this regard, it is noteworthy that visual cortex hyperexcitability (as measured by the pattern glare task) was related to lifetime occurrence out-of-body experiences [25,26], a form of altered states of consciousness that might be associated with absorption [29,68].

Possible brain areas that may be involved in visual snow experiences are the associative visual cortex and the thalamus [5]. A possible next step could be to study the general excitability of the associative visual cortex and the activity of the thalamus in participants experiencing visual snow and showing a greater capacity to be engrossed in visual experiences and imagination. Visual receptors and neurons demonstrate continuous activity with or without sensory information on the retinae. Neural activity in visual areas without sensory stimulation is typically labeled visual noise [69]. Such lower-level noise activity may be suppressed by certain brain areas, possibly including the thalamus, and not enter awareness. One possible explanation for visual snow experiences may be less suppression or lower consciousness thresholds in observers with hyperexcitability of the visual cortex.

## Further directions for future research

Our findings were obtained in a nonclinical sample. It is presently unclear whether similar findings would be obtained in clinical groups with visual snow. Studies with clinical groups using methodologies similar to ours would be needed to clarify this question. We found converging evidence with clinical studies relatively to associations of visual snow with entoptic

phenomena and tinnitus, and relatively to progression and origin of visual snow, but, to the best of our knowledge, the association with absorbed states is reported here for the first time. Future studies are needed to confirm this correlation in clinical groups and in groups of the general population. Online surveys allow the collection of large samples, but have the limitation of reliance on self-report. In the future, it would be useful that survey studies are complemented by qualitative data based on interviews and by test-retest studies to assess the coherence of the responses across a short period of time.

We cannot entirely rule out that some participants are more prone to respond affirmatively to questions regarding borderline sensations motivated by social desirability. However, several studies reveal that people scoring higher in absorption, as measured by the Tellegen Absorption Scale, do not tend to give more socially desirable responses [70–73]; if anything, reports of higher absorption have been related to lower social desirability responding [71–73]. The average scores of the MODTAS in the present study ranged from 63.64 to 66.06 ($25.89 \leq SD \leq 27.33$), which are similar to those obtained in other studies [37,74], suggesting that the present sample had no particular bias to acquiescence. Still, future research should address this question not only by including measures of social desirability, but also measures of various borderline sensations that are not expected to correlate with absorption (and visual snow) in order to test differential associations. Although we should expect that absorption mediates an association between visual snow and many altered states of consciousness, there is no reason to expect that visual snow would correlate with borderline sensations including flow states in activities that require goal-directed attention (e.g., in work or sports) [70,75], states of higher mindful attention [61], or otherwise exceptional states of consciousness that may result from goal-directed attentional control [28,61].

Future research would also benefit from using psychophysical tasks that are indicators of visual cortex hyperexcitability and that were shown to differentiate patients with visual snow syndrome from controls (tasks of center-surround contrast suppression and luminance increment detection threshold) [11], and migraine patients from controls (pattern-glare task) [26].

## Final remarks

Visual snow seems to be a relatively common phenomenon with many people experiencing it always or almost always. Many people are not distressed, implying that it is not a distressing phenomenon *per se*. In the nonclinical group in the present investigation, we found visual snow correlates similar to those that have already been observed in clinical groups: tinnitus and entoptic phenomena. We also confirmed that visual snow is associated with a greater capacity to be attentionally absorbed, i.e., the capacity to be fascinated.

Could clinical cases be characterized by more intense visual snow? It is possible, but difficult to confirm. Visual snow is an inherently subjective experience. An intensity that is comfortable for one person could be excessive and distressful for another. If one is anxious about the experience of visual snow or desires to conform to what is perceived to be "normal visual perception", this person might seek clinical advice. These aspects deserve to be addressed by clinicians. Cognitive-behavioral therapies might offer a possible intervention to reduce distress related to visual snow. In some cases, reassuring distressed people that visual snow can be a normal experience may already be an effective intervention.

There exist anecdotal reports of people who talked to their parents about the visual snow when they were children and were shocked by their parents' lack of understanding (e.g. [76]). Parental education in this regard seems important. As shown in Table 7, one respondent interpreted visual snow as terrifying witches when she was a child. It is important to raise attention to visual snow in pediatric care.

An important question still needs be asked. If there are so many people who are not distressed and even enjoy visual snow, why has scientific attention been so focused on people who are distressed? A possible answer lies in the difficulties that many people have disclosing their subjective experiences, because of fearing being misunderstood. An example is provided by a study of women with visual synesthesias during sexual activity, i.e., women who see colorful forms while experiencing sexual desire, arousal, or orgasm. These experiences are pleasant and enriching and are associated with greater sensory and emotional involvement (absorption) in sexual intercourse. Nevertheless, many of these women felt dissatisfied because they had difficulties sharing their synesthetic experiences with their partners [77]. Perhaps it is time to be more open and accepting of these possibly fascinating and intriguing aspects of human experience. Increasing awareness of visual snow by talking about it is one important step. To extend understanding to many other intriguing phenomena in visual or other sensory modalities, experienced as pleasant or not, will be another step. Conducting research like ours could help foster awareness of these phenomena.

More research is needed to establish the prevalence of visual snow and its underlying neurophysiological determinants, to better understand its psychological and behavioral correlates, and to characterize in greater detail emotional reactions to it.

## Supporting information

**S1 Table. Descriptive statistics for low blood pressure, blood pressure drops, ophthalmological problems, computer screen time, and lifetime use of psychoactive substances.**
(DOCX)

**S2 Table. Frequency of tinnitus.**
(DOCX)

**S3 Table. Frequency of floaters and blue field entoptic phenomena.**
(DOCX)

**S4 Table. Pearson's correlations between absorption and visual snow (dichotomously measured as never experiencing it vs. at least 10% of the time experiencing it).**
(DOCX)

**S5 Table. Factor loadings of the items of the Modified Tellegen Absorption Scale (MOD-TAS).**
(DOCX)

## Author Contributions

**Conceptualization:** Rui Miguel Costa.

**Data curation:** Rui Miguel Costa, Pedro Campos.

**Formal analysis:** Rui Miguel Costa.

**Investigation:** Rui Miguel Costa, Pedro Campos, Madalena Wiborg, Catarina Rebôlo.

**Methodology:** Rui Miguel Costa.

**Project administration:** Rui Miguel Costa.

**Resources:** Rui Miguel Costa.

**Supervision:** Rui Miguel Costa.

**Validation:** Rui Miguel Costa.

**Visualization:** Rui Miguel Costa, Jürgen Kornmeier.

**Writing – original draft:** Rui Miguel Costa.

**Writing – review & editing:** Marc Wittmann, Jürgen Kornmeier.

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
