## [Decision Letter · Decision Letter 0]

26 Apr 2022

PONE-D-21-32390Prevalence of visual snow and relation to attentional absorptionPLOS ONE

Dear Dr. Costa,

Thank you for submitting your manuscript to PLOS ONE. After careful consideration, we feel that it has merit but does not fully meet PLOS ONE’s publication criteria as it currently stands. Therefore, we invite you to submit a revised version of the manuscript that addresses the points raised during the review process. Among other comments, both reviewers raise the point that the study is based purely on self-report data and suggest further analyses. All comments should be addressed in a major revision.

We look forward to receiving your revised manuscript.

Kind regards,

Guido Hesselmann

Academic Editor

PLOS ONE

Journal Requirements:

Reviewers' comments:

Reviewer's Responses to Questions

**Comments to the Author**

1. Is the manuscript technically sound, and do the data support the conclusions?

Reviewer #1: Partly

Reviewer #2: Partly

2. Has the statistical analysis been performed appropriately and rigorously? 

Reviewer #1: No

Reviewer #2: Yes

3. Have the authors made all data underlying the findings in their manuscript fully available?

Reviewer #1: No

Reviewer #2: Yes

4. Is the manuscript presented in an intelligible fashion and written in standard English?

Reviewer #1: Yes

Reviewer #2: Yes

5. Review Comments to the Author

Reviewer #1: Visual snow was first described in 1995 by Liu et al and in the 27 years since then more than 100 papers on the condition have been published. The publication rate has been increasing year on year, and in 2021 35 papers were published on the subject. Few of these papers have attempted to explore the prevalence and nature of the condition as it might exist in the general population, and this is what the authors have sought to do. Their efforts may be likened to those of Braithwaite et al 2013 who studied out of body experiences (close to “ESP” in the authors’ absorbance scale) in the general population and found a high prevalence, related to results of visual tests. The danger with efforts of this kind is that in taking the reports of healthy participants at face value, there is an implicit assumption that the phenomena as reported are indeed similar to those of clinical patients. In order to demonstrate the similarity of reported phenomena, it might have been instructive to have used the questionnaire tools with clinical patients experiencing diagnosed visual stress. That said, the results are in line with earlier studies of this kind, and are interesting in giving us an idea of the prevalence of reports of “visual snow”-like phenomena and in relating these to questionnaire measures of absorption (the propensity to be engrossed in sensory and imaginary experiences). Again, there are dangers that spuriously high measures of a relationship between these two variables, were obtained because of acquiescence of the participant. Can the authors show that willingness to admit to borderline sensations was not responsible for the relationship between visual snow and absorption? Are the questionnaires effectively measuring the same thing (acquiescence) twice?

The extensive literature on visual stress seems to have converged on the idea that the phenomenon is attributable to a hyperexcitability of the visual cortex. There are simple psychophysical correlates of this hyperexcitability that could have been used in this study. For example, McKendrick et al (2017) demonstrated reduced center-surround contrast suppression (p = 0.03) and elevated luminance increment thresholds in noise (p = 0.02). Braithwaite et al (2013) used the Pattern Glare test, which shows elevated scores in migraine. Such tests are perhaps less likely to be influenced by acquiescence.

This willingness to report borderline sensations assumes importance when one considers that nearly half of the sample reported experiencing visual snow, at least some of the time. Is the relationship with absorption obtained with such a population likely to apply when stricter diagnostic measures are used? Was the correlation with absorbance dependent on the frequency with which visual snow was experienced?

Study 1 demonstrated that visual snow frequency correlated directly with the frequency

of two entoptic phenomena: floaters and blue field entoptic phenomenon (see Table 4). It would be nice to know whether this is acquiescence on the part of the observer. Were there other reports with which this may be contrasted?

“An untested possibility is that people experiencing visual snow may be more engrossed in perceptual and imaginary experiences, because visual snow seems to result from hyperexcitability of the visual cortex”. The authors appear to be suggesting that hyperexcitability of the visual cortex necessarily gives rise to perceptual and imaginary experiences. There is little hard evidence for such a view. For example, individuals with photosensitive epilepsy are not more likely than anyone else to experience “perceptual and imaginary experiences”, although they do suffer from visual discomfort. If the authors wish to take this conceptual leap, perhaps they could expand upon the logical steps and provide the evidence for each.

The description of the methods could be increased in detail. How was the question concerning tinnitus worded? Did it refer to “ringing/buzzing in the ears” or simply “tinnitus”.

It would be useful to know how the various aspects of absorption (imaginative involvement, aesthetic involvement, altered states of consciousness, synaesthesia and ESP) correlate one with another. Was there a single factor (absorption) or were the responses more multidimensional, as Table 4 suggests? If multidimensional, this would weaken the thesis presented in this paper.

“The prevalence of those experiencing visual snow was remarkably similar regardless of whether the assessment was performed with the aid of animated graphic simulations or participants were simply asked about the frequency of seeing “dots of light”.” This statement appears at variance with the data: 14% of the sample reported seeing visual snow continuously when the representation was visual, but only 2.6 and 1.4% when the representation was verbal (Table 2). As the authors themselves point out “the graphic simulation may have been more effective in calling attention to the fact that visual snow is “permanently or usually there”.” Curiously, Table 4 indicates that the correlation between visual snow and “distress caused by visual snow” was greater in Study 1 than in Study 2, whereas in Study 2 there was greater ”fascination with visual snow”. This suggests that the two studies were measuring different aspects of perception, perhaps because of the manner in which the perception was represented.

“…results … indicate that visual snow is not an all-or-nothing phenomenon, i.e., it is not permanently present in the visual field of those who experience it.” This is an important observation. In this reviewer’s experience the visual snow is easier to treat (with spectral filters) when the perception is labile and dependent on the visual scene than when more permanent.

Can the authors segregate their respondents into those who experience migraine with aura and those who do not? They might then find a stronger relationship with visual snow than with migraine overall. The prevalence of migraine in their population is more than twice that expected; this suggests that the “migraine” reported would not conform to conventional criteria.

“Visual snow seems to be a relatively common phenomenon with many people experiencing it always or nearly always. Many people are not distressed, implying that it is not a distressing phenomenon per se.” The phenomenon is likely to be more distressing the more frequently it is experienced, and particularly if it is invariably present, invading close visual work. Was there a relationship between distress and the frequency with which the phenomenon was experienced? If not, why not?

The paragraph concerning 5HT receptors appears, to me at least, unwarranted speculation, particularly given the other more mundane explanations for the correlation between visual snow and absorption.

References

Braithwaite, J.J., Broglia, E., Brincat, O., Stapley, L. Wilkins, A.J., Takahashi, C. (2013) Signs of increased cortical hyperexcitability selectively associated with spontaneous anomalous bodily experiences in a non-clinical population. Cognitive Neuropsychiatry, http://dx.doi.org/10.1080/13546805.2013.768176]

AllisonM. McKendrick, YuMan Chan, Melissa Tien, Lynette Millist, Meaghan Clough, Heather Mack, Joanne Fielding, Owen B. White. Behavioral measures of cortical hyperexcitability assessed in people who experience visual snow. Neurology Mar 2017, 88 (13) 1243-1249;

DOI: 10.1212/WNL.0000000000003784

Reviewer #2: This manuscript addresses an important topic, and present a broad data set from a large cohort in order to draw its conclusions.

My main concern with the study is that it is based purely on self-report data. As such, the question arises as to the extent to which the results present meaningful variation in the dimensions of interest. For example, it seems almost certain that some participants will simply more inclined to agree with the types of statements included. From this, a few more specific queries:

How could (or was) this controlled for - that is, how do we know that the responses actually indicated (e.g.) experience of visual snow rather than tendency to provide positive responses. Broadly, the extent to which we are able to confidently make use of the data depend on the extent to which we can trust the responses as meaningful, so it is important that this is understood, and at the least discussed. Ideally, we ought to be provided with some reasons to be confident on this issue.

There are some aspects of this where comparisons can be made, as there are items for which expected values are available. For example, we know that prevalence of migraine, and of tinnitus. To what extent do the result agree with these?

In the case of migraine, the response seem to be at the top end (or higher) than expected values. While these do differ, consensus is that we would expect about 10 (men) to 15 (women) percent. The results reported are a lot higher, how can this be reconciled? Also do you see the expected sex differences here? Can similar comparisons with expected norms be made with other measures?

Again sticking with migraine, I see problem here in that this is left undefined, so that participants would need to know whether they had migraine or not. This could lead to under-reporting (ie relying on having a formal diagnosis) or over-reporting (inaccurately attributing headaches as migraine). In the absence of formal diagnoses, it would have been possible to include a brief set of questions targeted at the International Headache Society criteria, or some such.

6. PLOS authors have the option to publish the peer review history of their article (what does this mean?). If published, this will include your full peer review and any attached files.

Reviewer #1: **Yes: **Arnold Wilkins

Reviewer #2: No

---

## [Author Response · Author response to Decision Letter 0]

17 Jul 2022

Response to Reviewers

Response to Reviewer 1

>>>>> Reviewer #1: Visual snow was first described in 1995 by Liu et al and in the 27 years since then more than 100 papers on the condition have been published. The publication rate has been increasing year on year, and in 2021 35 papers were published on the subject. Few of these papers have attempted to explore the prevalence and nature of the condition as it might exist in the general population, and this is what the authors have sought to do. 

R: We thank the Reviewer for his thoughtful comments.

>>>>> Reviewer #1: Their efforts may be likened to those of Braithwaite et al 2013 who studied out of body experiences (close to “ESP” in the authors’ absorbance scale) in the general population and found a high prevalence, related to results of visual tests.

R: That is an important observation. Now, when we discuss the possibility of visual snow being associated with altered states of consciousness by means of visual cortex hyperexcitability by adding to the “Absorption and visual snow” subsection that “In this regard, it is noteworthy that visual cortex hyperexcitability (as measured by the pattern glare task) was related to lifetime occurrence out-of-body experiences [55], a form of altered states of consciousness that might be associated with absorption [22,56]”.

We added the reference:

55. Braithwaite JJ, Broglia E, Brincat O, Stapley L, Wilkins AJ, Takahashi C. (2013) Signs of increased cortical hyperexcitability selectively associated with spontaneous anomalous bodily experiences in a non-clinical population. Cogn Neuropsychiatry 2013; 18: 549-573. 

>>>>> Reviewer #1: The danger with efforts of this kind is that in taking the reports of healthy participants at face value, there is an implicit assumption that the phenomena as reported are indeed similar to those of clinical patients. In order to demonstrate the similarity of reported phenomena, it might have been instructive to have used the questionnaire tools with clinical patients experiencing diagnosed visual stress.

R: In the new subsection “Further directions of future research”, we highlighted that “Our findings were obtained in a nonclinical sample. It is presently unclear whether similar findings would be obtained in clinical groups with visual snow. Studies with clinical groups using methodologies similar to ours would be needed to clarify this question. We found converging evidence with clinical studies relative to associations of visual snow with entoptic phenomena, tinnitus, migraine, and progression and origin of visual snow, but, to the best of our knowledge, the association with absorbed states is reported here for the first time. Future studies are needed to confirm this correlation in clinical groups and in groups of the general population”.

>>>>> Reviewer #1: That said, the results are in line with earlier studies of this kind, and are interesting in giving us an idea of the prevalence of reports of “visual snow”-like phenomena and in relating these to questionnaire measures of absorption (the propensity to be engrossed in sensory and imaginary experiences). Again, there are dangers that spuriously high measures of a relationship between these two variables, were obtained because of acquiescence of the participant. Can the authors show that willingness to admit to borderline sensations was not responsible for the relationship between visual snow and absorption? Are the questionnaires effectively measuring the same thing (acquiescence) twice?

R: We think that there are reasons supporting that the findings were not due to acquiescence, although we cannot entirely rule out that acquiescence may have played a role. This is a limitation that might be addressed in future studies, and we provide possible directions to do so. As noted above, we now highlight in the new subsection “Further directions for future research” that “We found converging evidence with clinical studies relative to associations of visual snow with entoptic phenomena, tinnitus, migraine, and progression and origin of visual snow”. 

Moreover, we added that “We cannot entirely rule out that some participants are more prone to respond affirmatively to questions regarding borderline sensations motivated by social desirability. However, several studies reveal that people scoring higher in absorption, as measured by the Tellegen Absorption Scale, do not tend to give more socially desirable responses [58-61]; if anything, reports of higher absorption have been related to lower social desirability responding [59-61]. The average scores of the MODTAS in the present study ranged from 63.64 to 66.06… (25.89 ≤ SD ≤ 27.33), which are similar to those obtained in other studies [30,62], suggesting that the present sample had no particular bias to acquiescence. Still, future research should address this question not only by including measures of social desirability, but also measures of various borderline sensations that are not expected to correlate with absorption (and visual snow) in order to test differential associations. Although we should expect that absorption mediates an association between visual snow and many altered states of consciousness, there is no reason to expect that visual snow would correlate with borderline sensations including flow states in activities that require goal-directed attention (e.g, in work or sports) [58,63,], states of higher mindful attention [64], or otherwise exceptional states of consciousness that may result from goal-directed attentional control [21,64]”. 

We added the following references:

58. Koehn S, Morris T, Watt AP. Correlates of dispositional and state flow in tennis competition. J Appl Sport Psychol 2013; 25: 354-369.

59. Gick M, McLeod C, Hulihan D. Absorption, social desirability, and symptoms in a behavioral medicine population. J Nerv Ment Disease 1997; 185: 454-458.

60. Grady KE. The absorption scale: factor analytic assessment. Int J Clin Exp Hypnosis 1980; 28: 281-288.

61. Wickramasekera II IE. Empathic features of absorption and incongruence. Am J Clin Hypn 2007; 50: 59-69.

62. Scacchia P, De Pascalis V. Effects of prehypnotic instructions on hypnotizability and relations between hypnotizability, absorption, and empathy. Am J Clin Hypn 2020; 62: 231-266.

63. Marty-Dougas J, Smilek D. Deep, effortless concentration: re-examining the flow concept and exploring relations with inattention, absorption, and personality. Psychol Res 2019; 83: 1760-1777.

64. Leppanen, M. L., & Kim, K. Absorption and mindfulness reflect distinct patterns of attentional control and self-related processing. J Indiv Diff 2021. https://doi.org/10.1027/1614-0001/a000360

>>>>> Reviewer #1: The extensive literature on visual stress seems to have converged on the idea that the phenomenon is attributable to a hyperexcitability of the visual cortex. There are simple psychophysical correlates of this hyperexcitability that could have been used in this study. For example, McKendrick et al (2017) demonstrated reduced center-surround contrast suppression (p = 0.03) and elevated luminance increment thresholds in noise (p = 0.02). Braithwaite et al (2013) used the Pattern Glare test, which shows elevated scores in migraine. Such tests are perhaps less likely to be influenced by acquiescence.

R: We added to the subsection “Further directions for future research” that “Future research would also benefit from using psychophysical tasks that are indicators of visual cortex hyperexcitability and that were shown to differentiate patients with visual snow syndrome from controls (tasks of center-surround contrast suppression and luminance increment detection threshold) [11], and migraine patients from controls (pattern-glare task)”.

As noted above, we added the reference to Braihwaite and colleagues (2013). We had already included the refence to McKendrick and colleagues (2017).

>>>>> Reviewer #1: This willingness to report borderline sensations assumes importance when one considers that nearly half of the sample reported experiencing visual snow, at least some of the time. Is the relationship with absorption obtained with such a population likely to apply when stricter diagnostic measures are used? 

R: As stated above, we added that “Our findings were obtained in a nonclinical sample. It is presently unclear whether similar findings would be obtained in clinical groups with visual snow. Studies with clinical groups using methodologies similar to ours would be needed to clarify this question. We found converging evidence with clinical studies relative to associations of visual snow with entoptic phenomena, tinnitus, migraine, and progression and origin of visual snow, but, to the best of our knowledge, the association with absorbed states is reported here for the first time. Future studies are needed to confirm this correlation in clinical groups and in groups of the general population”.

>>>>> Reviewer #1: Was the correlation with absorbance dependent on the frequency with which visual snow was experienced?

R: Yes, and we discuss this finding. We added to the subsection “Visual snow and absorption” that “These correlations were largely explained by persons who never see visual snow having lower absorption in comparison with those who experience it at least 10% of the time. When the analyses were restricted to the subgroup seeing visual snow at least 10% of the time, the correlations between visual snow and absorption (and absorption dimensions) became nonsignificant or significant with a reduced effect size.

Point-biserial correlations were run between absorption and visual snow (dichotomously measured as no visual snow vs. seeing visual snow at least 10% of the time). With this approach, visual snow was always significantly associated with absorption (and absorption dimensions): Study 1: .14 ≤ r ≤ .26, all p ≤ .004; Study 2: .19 ≤ r ≤ .32, all p ≤ .001; Study 3: .26 ≤ r ≤ .37, all p < .001 (see Table D in the Appendix). In these point-biserial correlations with visual snow dichotomously considered), altered states of consciousness continued to be the dimension of absorption most strongly correlated with visual snow. Thus, absorbed states do not seem to be associated with persistent visual snow, but rather with some susceptibility to experience it”.

>>>>> Reviewer #1: Study 1 demonstrated that visual snow frequency correlated directly with the frequency

of two entoptic phenomena: floaters and blue field entoptic phenomenon (see Table 4). It would be nice to know whether this is acquiescence on the part of the observer. Were there other reports with which this may be contrasted?

R: It is unlikely that these correlations are solely due to acquiescence. To clarify this, we re-elaborated the subsection “Visual snow and entoptic phenomena”. This subsection now reads as following: “Study 1 demonstrated that visual snow frequency correlated directly with the frequency of two entoptic phenomena: floaters and blue field entoptic phenomenon (see Table 4). This coincides with research in clinical samples in which visual snow and entoptic phenomena often occur together. Schankin and colleagues observed that blue field entoptic phenomena occurred in 79% of a group of patients with visual snow; floaters occurred in 81% [2]. In fact, entoptic phenomena are among the additional visual phenomena required to diagnose the visual snow syndrome [1-3]. Even in nonclinical groups, more people apparently report visual snow with additional visual phenomena than without [3,18]”.

In addition, as stated above, we recognized the possible influence of acquiescence in the discussion of limitations, and ways to address this issue in future studies.

>>>>> Reviewer #1: An untested possibility is that people experiencing visual snow may be more engrossed in perceptual and imaginary experiences, because visual snow seems to result from hyperexcitability of the visual cortex”. The authors appear to be suggesting that hyperexcitability of the visual cortex necessarily gives rise to perceptual and imaginary experiences. There is little hard evidence for such a view. For example, individuals with photosensitive epilepsy are not more likely than anyone else to experience “perceptual and imaginary experiences”, although they do suffer from visual discomfort. If the authors wish to take this conceptual leap, perhaps they could expand upon the logical steps and provide the evidence for each.

R: We re-elaborated this part, which now reads as following: “An untested possibility is that people experiencing visual snow may be more engrossed in perceptual and imaginary experiences, because visual snow may result from hyperexcitability of the visual cortex [4-12]. If that is the case, such hyperexcitability could intensify visual perceptions and mental imagery, thereby facilitating attentional absorption in percepts and mental images. A core characteristic of engrossment in mental imagery is imagination vividness [20-22], and there are reasons to think that visual snow may be associated with imagination vividness. Visual snow has also been associated with greater activity [5-10] and greater gray matter volume of the lingual gyrus [8,15], an area of the associative visual cortex that tends to become more active when more vivid mental images are generated or when more vivid memories are recalled [23-25]. Also, the lingual gyrus was found to be larger in people with greater ability to imagine [26], and more active during tasks of internally directed attention [27] and mental imagery [28]. Such increased activity of the visual cortex is thought to facilitate the awareness of visual snow, which might result from the conscious perceptions of a particular form of sensory information that normally remains subthreshold [5]. There is less evidence suggesting a link between visual cortex hyperactivity and intensified perception of external events, but engrossment in sensory perceptions is often accompanied by self-generated imagination [21,22]. In addition, a small study reported that people visual snow have intensified sensory experiences in the olfactory and tactile modalities [29]”.

We added the following references:

23. Belardinelli MO, Palmiero M, Sestieri C, Nardo D, Di Matteo R, Londei A, D’Ausilio A, Ferretti A, Del Grata C, Romani CL. An fMRI investigation on image generation in different sensory modalities: the influence of vividness. Acta Psychol 2009; 132: 190-200.

24. Gilboa A, Winocur G, Grady CL, Hevenor SJ, Moscovitch M. Remembering our past: functional neuroanatomy of recollection of recent and very remote personal events. Cereb Cortex 2004; 14: 1214-1225.

25. Sheldon S, Levine B. Same as it ever was: Vividness modulates the similarities and differences between the neural networks that support retrieving remote and recent autobiographical memories. Neuroimage 2013; 83: 880-991.

>>>>> Reviewer #1: The description of the methods could be increased in detail. How was the question concerning tinnitus worded? Did it refer to “ringing/buzzing in the ears” or simply “tinnitus”.

R: As we noted in the “Measures” subsection, “The frequency of tinnitus was assessed by asking participants about the amount of time they hear a ringing, buzzing, hissing, or other sounds that do not come from the external world”.

>>>>> Reviewer #1: It would be useful to know how the various aspects of absorption (imaginative involvement, aesthetic involvement, altered states of consciousness, synaesthesia and ESP) correlate one with another. Was there a single factor (absorption) or were the responses more multidimensional, as Table 4 suggests? If multidimensional, this would weaken the thesis presented in this paper.

R: Yes, there is a single factor. We clarify this issue by doing a series of new statistical analyses. Hence, we added to subsection “Visual snow and absorption” that “Across the three studies, the five dimensions of the MODTAS were strongly correlated with each other (.45 ≤ r ≤ .74, all p < .001). After collapsing the MODTAS data from the three studies, Cronbach’s alpha for the total scale was .95. Acceptable results were obtained in an exploratory factor analysis using the principal components method with varimax rotation, carried with the 34 items of the MODTAS and requesting the extraction of one single factor. This explained 36.85% of variance (KMO = .97; Bartlett’s Test of Sphericity: approximate χ2 = 6352.84, df = 561, p < .001). For 33 items, loadings ranged between .50 and .72. Item 2 had a loading of .40. If these analyses were done separately for the three studies, results were similar (see Table E in the Appendix). This indicates that the MODTAS items express a common latent variable: engrossment in imaginary and perceptual experiences with this kind of immersion in perceptual experiences being a rather passive and receptive experience, largely involuntary, like in states of fascination and wonder [20-22]”.

>>>>> Reviewer #1: “The prevalence of those experiencing visual snow was remarkably similar regardless of whether the assessment was performed with the aid of animated graphic simulations or participants were simply asked about the frequency of seeing “dots of light”.” This statement appears at variance with the data: 14% of the sample reported seeing visual snow continuously when the representation was visual, but only 2.6 and 1.4% when the representation was verbal (Table 2). 

R: We corrected that sentence, which now reads as “The prevalence of those who never see visual snow was remarkably similar regardless of whether the assessment was performed with the aid of animated graphic simulations or participants were simply asked about the frequency of seeing “dots of light” (roughly 55%-56%). This means that roughly 45% of the respondents reported some visual experience of snow with varying degrees of frequency”.

>>>>> Reviewer #1: As the authors themselves point out “the graphic simulation may have been more effective in calling attention to the fact that visual snow is “permanently or usually there”.” Curiously, Table 4 indicates that the correlation between visual snow and “distress caused by visual snow” was greater in Study 1 than in Study 2, whereas in Study 2 there was greater ”fascination with visual snow”. This suggests that the two studies were measuring different aspects of perception, perhaps because of the manner in which the perception was represented.

R: Thanks for drawing attention to that. There was a mistake in Table 4 that was corrected. In both studies, distress correlates significantly with visual snow frequency, albeit the correlation in Study 1 is in fact larger than in Study 2. The correlation with fascination was indeed larger in Study 2 (we provide details on this question in replying to another concern below). To discuss these findings, we added to the subsection “Reactions of visual snow and tinnitus” that “More research is needed to clarify how the frequency of visual correlates with psychological reactions to it. We cannot exclude that the discrepancy in Study 1 and 2 is due to some bias in sample selection or to different aspects of visual snow being measured, as the studies used different forms of assessment”.

>>>>> Reviewer #1: “…results … indicate that visual snow is not an all-or-nothing phenomenon, i.e., it is not permanently present in the visual field of those who experience it.” This is an important observation. In this reviewer’s experience the visual snow is easier to treat (with spectral filters) when the perception is labile and dependent on the visual scene than when more permanent.

R: Thank you.

>>>>> Reviewer #1: Can the authors segregate their respondents into those who experience migraine with aura and those who do not? They might then find a stronger relationship with visual snow than with migraine overall. The prevalence of migraine in their population is more than twice that expected; this suggests that the “migraine” reported would not conform to conventional criteria.

R: Unfortunately, we cannot. We simply asked participants if they had migraine. It is possible that many of them misjudged their headaches as migraines. We noted this as a limitation, and re-elaborated the discussion on our findings relative to migraine, which also addresses a similar concern posed by Reviewer 2. That part in the subsection “Visual snow and migraine” now reads as following: “Table 4 shows weak but significant associations between migraine and visual snow frequency, whether visual snow was measured with animated graphic simulations or by mere verbal description. This coincides with studies in clinical samples showing that migraine is a correlate of visual snow, but the small effects might be due to many respondents misperceiving their headaches as migraines, since we asked about “migraine” without further specifications This is suggested by the large proportion of respondents reporting to have migraines. However, in both Study 1 and 2, we found a greater proportion of women reporting migraine with an expected sex ratio [45]: 25% vs. 15% in Study 1, and 37% vs. 23% in Study 2. The study with a British representative sample [18] demonstrated a marginal association between the visual snow syndrome and headache, which disappeared in a multiple regression in which tinnitus was the only significant predictor. But as noted, participants in the present study were simply asked about headaches. Future research would gain from applying stricter established criteria to define migraine (e.g., those of the International Headache Society)”.

We added the following reference:

45. Buse DC, Loder EW, Gorman JA, Stewart WF, Reed ML, Fanning KM, Serrano D, Lipton RB. Sex differences in the prevalence, symptoms, and associated features of the American Migraine Prevalence and Prevention (AMPP) Study. Headache 2013; 53: 1278-1299.

>>>>> Reviewer #1: “Visual snow seems to be a relatively common phenomenon with many people experiencing it always or nearly always. Many people are not distressed, implying that it is not a distressing phenomenon per se.” The phenomenon is likely to be more distressing the more frequently it is experienced, and particularly if it is invariably present, invading close visual work. Was there a relationship between distress and the frequency with which the phenomenon was experienced? If not, why not?

R: Yes, we now give more details on this issue. We noted in subsection “Reactions to visual snow and tinnitus” that “As shown in Table 4, visual snow frequency correlated with greater associated distress in Studies 1 and 2, but more strongly so in Study 1. Visual snow frequency correlated with fascination moderately in Study 2, and was uncorrelated in Study 1. It seems that greater frequency of visual snow may contribute to distress in some people, but many others do not feel uncomfortable with the persistent experience. In a multiple regression done with the Study 2 sample, we found that distress (β = .24, p = .004) and fascination (β = .36, p < .001) were directly and independently associated with visual snow frequency. To understand why some people are distressed, and others are not is a crucial one. Comparing groups of people with and without distress might bring clarification to this important question”.

>>>>> Reviewer #1: The paragraph concerning 5HT receptors appears, to me at least, unwarranted speculation, particularly given the other more mundane explanations for the correlation between visual snow and absorption.

R: We deleted the paragraph.

Response to Reviewer 2

>>>>> Reviewer #2: This manuscript addresses an important topic, and present a broad data set from a large cohort in order to draw its conclusions.

R: Thank you.

>>>>> Reviewer #2: My main concern with the study is that it is based purely on self-report data. As such, the question arises as to the extent to which the results present meaningful variation in the dimensions of interest. For example, it seems almost certain that some participants will simply more inclined to agree with the types of statements included. From this, a few more specific queries:

How could (or was) this controlled for - that is, how do we know that the responses actually indicated (e.g.) experience of visual snow rather than tendency to provide positive responses. Broadly, the extent to which we are able to confidently make use of the data depend on the extent to which we can trust the responses as meaningful, so it is important that this is understood, and at the least discussed. Ideally, we ought to be provided with some reasons to be confident on this issue.

R: Reviewer 1 raised a similar concern. Below, we provide a reply that address both Reviewers’ concerns in addition to a reply to more specific concerns of Reviewer 2.

>>>>> Reviewer #2: There are some aspects of this where comparisons can be made, as there are items for which expected values are available. For example, we know that prevalence of migraine, and of tinnitus. To what extent do the result agree with these? In the case of migraine, the response seem to be at the top end (or higher) than expected values. While these do differ, consensus is that we would expect about 10 (men) to 15 (women) percent. The results reported are a lot higher, how can this be reconciled? Also do you see the expected sex differences here? Can similar comparisons with expected norms be made with other measures?

R: We re-elaborated the subsection “Visual snow and migraine”. Now it can be read that “Table 4 shows weak but significant associations between migraine and visual snow frequency, whether visual snow was measured with animated graphic simulations or by mere verbal description. This coincides with studies in clinical samples showing that migraine is a correlate of visual snow, but the small effects might be due to many respondents misperceiving their headaches as migraines, since we asked about “migraine” without further specifications This is suggested by the large proportion of respondents reporting to have migraines. However, in both Study 1 and 2, we found a greater proportion of women reporting migraine with an expected sex ratio [45]: 25% vs. 15% in Study 1, and 37% vs. 23% in Study 2. The study with a British representative sample [18] demonstrated a marginal association between the visual snow syndrome and headache, which disappeared in a multiple regression in which tinnitus was the only significant predictor, but as noted, participants were simply asked about headaches. Future research would gain from applying stricter established criteria (e.g., those of the International Headache Society)”.

We added the reference:

45. Buse DC, Loder EW, Gorman JA, Stewart WF, Reed ML, Fanning KM, Serrano D, Lipton RB. Sex differences in the prevalence, symptoms, and associated features of the American Migraine Prevalence and Prevention (AMPP) Study. Headache 2013; 53: 1278-1299.

We wrote in the subsection “Visual snow and tinnitus” that “Tinnitus was reported to be experienced at least 10% of the time by 58% of respondents in Study 1 and 48% in Study 2 (see Figure 2). This is larger than the reported prevalence of tinnitus that ranges between 5% and 43% [42], but our measure is likely less conservative, as it could include persons who briefly hear sounds that do not disturb them. The proportion of respondents experiencing tinnitus more than 10% of time was 40% and 37% (see Figure 2), which is comparable to that of the literature [42]”.

We added the reference:

42. McCormack, A., Edmonson-Jones M, Somerset S, Hall D. A systematic review of the reporting of tinnitus prevalence and severity. Hear Res 2016; 337: 70-79.

We added to the subsection “Reactions to visual snow and tinnitus” that “For comparison with reported prevalence of bothersome tinnitus, we calculated the proportion of respondents reporting moderate to severe distress with tinnitus from the total sample (including those without tinnitus). We obtained a proportion of 17% in Study 1 and 20% in Study 2. The prevalence of bothersome tinnitus reported in the literature ranges from 3% to 30% [42]”.

We re-elaborated the subsection “Visual snow and migraine”. Now it can be read that “Table 4 shows weak but significant associations between migraine and visual snow frequency, whether visual snow was measured with animated graphic simulations or by mere verbal description. This coincides with studies in clinical samples showing that migraine is a correlate of visual snow, but the small effects might be due to many respondents misperceiving their headaches as migraines, since we asked about “migraine” without further specifications This is suggested by the large proportion of respondents reporting to have migraines. However, in both Study 1 and 2, we found a greater proportion of women reporting migraine with an expected sex ratio [42]: 25% vs. 15% in Study 1, and 37% vs. 23% in Study 2. The study with a British representative sample [18] demonstrated a marginal association between the visual snow syndrome and headache, which disappeared in a multiple regression in which tinnitus was the only significant predictor, but as noted, participants were simply asked about headaches. Future research would gain from applying stricter established criteria to define migraine (e.g., those of the International Headache Society)”.

We now highlight in the new subsection “Further directions of future research” that “We found converging evidence with clinical studies relative to associations of visual snow with entoptic phenomena, tinnitus, migraine, and progression and origin of visual snow”. 

Moreover, added to the same subsection that “We cannot entirely rule out that some participants are more prone to respond affirmatively to questions regarding borderline sensations motivated by social desirability. However, several studies reveal that people scoring higher in absorption, as measured by the MODTAS, do not tend to give more socially desirable responses [58-61]; if anything, reports of higher absorption have been related to lower social desirability responding [59-61]. The average scores of the MODTAS in the present study ranged from 63.64 to 66.06… (25.89 ≤ SD ≤ 27.33), which are similar to those obtained in other studies [30,62], suggesting that the present sample had no particular bias to acquiescence. Still, future research should address this question not only by including measures of social desirability, but also measures of various borderline sensations that are not expected to correlate with absorption (and visual snow) in order to test differential associations. Although we should expect that absorption mediates an association between visual snow and many altered states of consciousness, there is no reason to expect that visual snow would correlate with borderline sensations including flow states in activities that require goal-directed attention (e.g, in work or sports) [58,63,], states of higher mindful attention [64], or otherwise exceptional states of consciousness that may result from goal-directed attentional control [21,64]”.

We re-elaborated the subsection “Visual snow and entoptic phenomena”. Now it can be read that that “Study 1 demonstrated that visual snow frequency correlated directly with the frequency of two entoptic phenomena: floaters and blue field entoptic phenomenon (see Table 4). This coincides with research in clinical samples in which visual snow and entoptic phenomena often occur together. Schankin and colleagues observed that blue field entoptic phenomena occurred in 79% of a group of patients with visual snow; floaters occurred in 81% [2]. In fact, entoptic phenomena are among the additional visual phenomena required to diagnose the visual snow syndrome [1-3]. Even in nonclinical groups, more people apparently report visual snow with additional visual phenomena than without [3,18]”.

>>>>> Reviewer #2: Again sticking with migraine, I see problem here in that this is left undefined, so that participants would need to know whether they had migraine or not. This could lead to under-reporting (ie relying on having a formal diagnosis) or over-reporting (inaccurately attributing headaches as migraine). In the absence of formal diagnoses, it would have been possible to include a brief set of questions targeted at the International Headache Society criteria, or some such.

R: We addressed this concern above when replying to a previous one.

---

## [Decision Letter · Decision Letter 1]

24 Aug 2022

PONE-D-21-32390R1Prevalence of visual snow and relation to attentional absorptionPLOS ONE

Dear Dr. Costa,

Thank you for submitting your manuscript to PLOS ONE. After careful consideration, we feel that it has merit but does not fully meet PLOS ONE’s publication criteria as it currently stands. Therefore, we invite you to submit a revised version of the manuscript that addresses the points raised during the review process. The reviewers raise a couple of remaining points that should be addressed in a minor revision, including the hyperexcitability of the visual cortex as well as a more nuanced discussion of the study's limitations.

We look forward to receiving your revised manuscript.

Kind regards,

Guido Hesselmann

Academic Editor

PLOS ONE

Journal Requirements:

Reviewers' comments:

Reviewer's Responses to Questions

**Comments to the Author**

1. If the authors have adequately addressed your comments raised in a previous round of review and you feel that this manuscript is now acceptable for publication, you may indicate that here to bypass the “Comments to the Author” section, enter your conflict of interest statement in the “Confidential to Editor” section, and submit your "Accept" recommendation.

Reviewer #1: All comments have been addressed

Reviewer #2: (No Response)

2. Is the manuscript technically sound, and do the data support the conclusions?

Reviewer #1: Yes

Reviewer #2: Partly

3. Has the statistical analysis been performed appropriately and rigorously? 

Reviewer #1: Yes

Reviewer #2: Yes

4. Have the authors made all data underlying the findings in their manuscript fully available?

Reviewer #1: Yes

Reviewer #2: No

5. Is the manuscript presented in an intelligible fashion and written in standard English?

Reviewer #1: Yes

Reviewer #2: (No Response)

6. Review Comments to the Author

Reviewer #1: The manuscript reads well. The authors have considered carefully each of the points I raised and have altered the manuscript appropriately. I remain concerned that in the sentence "An untested possibility is that

people experiencing visual snow may be more engrossed in perceptual and imaginary experiences, because visual snow may result from hyperexcitability of the visual cortex [4-12]." Why should hyperexcitability of the visual cortex give rise to imaginary experiences? Perhaps the authors could elaborate their explanation with a sentence or two. There is in fact a little evidence of an association between hyperexcitability and imaginary experiences, as in the papers by Braithwaite et al, which the authors already cite.

Reviewer #2: I thank the authors for their revisions. As in the original manuscript, I have no issues with the data and analysis themselves, but remain cautious about what, if anything, can be concluded from these type of surveys.

I would ask therefore that these issues are put more strongly in the discussion.

On the point that participants may be simply inclined to agree with statements, this is mentioned rather briefly, then it is argued that it is probably not an issue due to agreement with past studies. I would like to see a more reflective statement ono the limitations of the current design as a way of providing independent evidence on this point .

On the comparison with more rigorous and accepted prevalence statistics, again the limitations of the current study do not really seem to be highlighted clearly enough. In the new section on page 23, this issues is acknowledge, but I am left wondering what, if anything, we can conclude from the responses on migraine. It would be very help to have a much more detailed discussion of the link to proper measures of the various phenomena and where the discrepancies arise .

On this point, and apologies for introducing another (albeit related) point, to what extent is it possible to differentiate different types of visual snow as entirely separate phenomena based on underlying mechanisms. Based purely on personal experience, I fully recognise the experience of visual snow resulting from sudden changes in blood pressure, but would not relate this to the experience of clinical visual snow, related to cortical excitability. A concern here is that the definition is very broad, allowing for a lot of positive responses.

Finally on the point of data availability - although a link has been I could not see the data, I think they have not been uploaded yet.

7. PLOS authors have the option to publish the peer review history of their article (what does this mean?). If published, this will include your full peer review and any attached files.

Reviewer #1: No

Reviewer #2: No

---

## [Author Response · Author response to Decision Letter 1]

16 Oct 2022

Response to Reviewers

Response to Reviewer #1

>>>>> The manuscript reads well. The authors have considered carefully each of the points I raised and have altered the manuscript appropriately. I remain concerned that in the sentence "An untested possibility is that people experiencing visual snow may be more engrossed in perceptual and imaginary experiences, because visual snow may result from hyperexcitability of the visual cortex [4-12]." Why should hyperexcitability of the visual cortex give rise to imaginary experiences? Perhaps the authors could elaborate their explanation with a sentence or two. There is in fact a little evidence of an association between hyperexcitability and imaginary experiences, as in the papers by Braithwaite et al, which the authors already cite.

R: We added three more references to support the claim that imagination vividness is related to visual cortex activity, and based on that we noted that if the visual cortex is hyperactive, it is likely that visual imagery is particularly vivid and attention-grabbing. To support this claim, we added more references, including the study by Braithwaite that was already cited in another part of the manuscript and another one from the same author. We added to the Introduction that “Studies show that vividness of mental imagery is related to activity of primary [21,22] and association visual cortex [22,23], suggesting that if visual cortex becomes hyperactive, imaginary experiences are likely to become more vivid, and as such more likely to grab attention, at least if attention was previously directed inwards. In accordance, pharmacologically-induced visual cortex hyperactivation through LSD correlates with increases in vividness of mental imagery, but not with other psychoactive effects [24]. Additionally, more responsiveness to the pattern glare task (an index of visual cortex hyperexcitability) was associated with history of out-of-body experiences, which can be seen as an extreme case of visual imagery vividness [25,26]. Pattern-glare task responsiveness was unrelated to history of sensed presence experiences, which are not characterized by vivid visual experiences, suggesting that the visual cortex hyperexcitability does not promote altered states of consciousness in general, but rather those characterized by strong visual imagery vividness [26]”.

Below we added that “Visual snow is associated is associated with hyperexcitability of the primary and association cortex [4-12], whose activity increases as visual imagery gets more vivid [21-24]”.

References 20 to 25 cited in these parts are the following:

21. Cui X, Jeter CB, Yang D, Montague PR, Eagleman DM. Vividness of mental imagery: individual variability can be measured objectively. Vision Res 2007; 47: 474-478.

22. Dijkstra N, Bosch SE, van Gerven MAJ. Vividness of Visual Imagery Depends on the Neural Overlap with Perception in Visual Areas. J Neurosci 2017; 37: 1367-1373.

23. Fulfold J, Milton F, Salas D, Smith A, Simler A, Winlove C, Zeman A. The neural correlates of visual imagery vividness – an fMRI study and literature review. Cortex 2018; 105: 26-40.

24. Carhart-Harris RL, Muthukumaraswamy S, Roseman L, Kaelen M, Droog W, Murphy K, Tagliazucchi E, Schenberg EE, Nest T, Orban C, Leech R, Williams LT, Williams TM, Bolstridge M, Sessa B, McGonigle J, Sereno MI, Nichols D, Hellyer PJ, Hobden P, Evans J, Singh KD, Wise RG, Curran HV, Feilding AF, Nutt DJ. Neural correlates of LSD experience revealed by multimodal neuroimaging. Proc Natl Acad Sci U S A 2016; 113: 4853-4858.

25. Braithwaite JJ, Broglia E, Bagshaw AP, Wilkins AJ. Evidence for elevated cortical hyperexcitability and its association with out-of-body experiences in the non-clinical population: New findings from a pattern-glare task. Cortex 2013; 49: 793-805.

26. Braithwaite JJ, Broglia E, Brincat O, Stapley L, Wilkins AJ, Takahashi C. (2013) Signs of increased cortical hyperexcitability selectively associated with spontaneous anomalous bodily experiences in a non-clinical population. Cogn Neuropsychiatry 2013; 18: 549-573.

Response to Reviewer #2

>>>>> I thank the authors for their revisions. As in the original manuscript, I have no issues with the data and analysis themselves, but remain cautious about what, if anything, can be concluded from these type of surveys. I would ask therefore that these issues are put more strongly in the discussion.

On the point that participants may be simply inclined to agree with statements, this is mentioned rather briefly, then it is argued that it is probably not an issue due to agreement with past studies. I would like to see a more reflective statement ono the limitations of the current design as a way of providing independent evidence on this point.

R: In the subsection “Further directions for future research”, we added that “surveys allow the collection of large samples, but have the limitation of reliance on self-report. In the future, it would be useful that survey studies are complemented by qualitative data based on interviews and by test-retest studies to assess the coherence of the responses across a short period of time”.

Meanwhile a new study came to our knowledge suggesting that the prevalence of frequent or constant visual snow is similar to the one obtained in the Study 1 of our manuscript. Because this is independent research that corroborates our findings, we now cited it. As such, we added to the Introduction that “A higher prevalence of visual snow was suggested by a study with 117 participants with history of hallucinogen consumption divided in three groups. The prevalence of those with frequent or constant visual snow varied roughly between 16% and 21% before use of hallucinogens, precluding the possibility of visual snow being related to Hallucinogen Persistent Perception Disorder (HPPD) by that time [20].”

Additionally, in the subsection “Prevalence of visual snow”, we added that “Notably, the prevalence of frequent or persistent visual snow we obtained in Study 1 is similar to that obtained in a study of users of hallucinogens before they started using this type of drugs [20], which rules out that hallucinogens were part of the etiology”.

We added the following reference:

20. Irvine A, Luke D. Apophenia, absorption and anxiety: Evidence for individual differences in positive and negative experiences of Hallucinogen Persisting Perceptual Disorder. J Psychedelic Stud 2022; 6: 88-103

>>>>> On the comparison with more rigorous and accepted prevalence statistics, again the limitations of the current study do not really seem to be highlighted clearly enough. In the new section on page 23, this issues is acknowledge, but I am left wondering what, if anything, we can conclude from the responses on migraine. It would be very help to have a much more detailed discussion of the link to proper measures of the various phenomena and where the discrepancies arise.

R: Even if migraine and visual snow were correlated in the expected direction, we acknowledge that the measure of migraine has limitations that may lead to overestimation of migraine cases. We deleted the analyses involving migraine and related discussion. In addition, we added more details when discussing our measures of tinnitus, entoptic phenomena, and absorption.

In the subsection “Visual snow and tinnitus”, we expanded the discussion on the possible limitations of our measure of tinnitus prevalence and how it compares with other measures employed to assess tinnitus prevalence. As such, we added that “However, there are no standardized measures for tinnitus, which makes prevalence estimates uncertain. Research on tinnitus prevalence has resorted to different types of questions across studies [49]. Although we cannot rule out that our measure of tinnitus overestimated its prevalence, it is not equally possible to rule out that other measures underestimated it, especially when it occurs briefly and/or infrequently. For example, sometimes questions focus on tinnitus (“noises in your head or ears”) for more than five minutes, which may not always happen. In contrast, our measure assesses a large spectrum of tinnitus frequencies without making reference to tinnitus duration, which includes very transient and infrequent experiences. In fact, when we do not count the participants reporting frequencies of 10% (see Figure 2 and Table B in Appendix), the prevalence is inside the reported range [49]. The proportion of respondents experiencing tinnitus more than 10% of time was 40% in Study 1 and 37% in Study 2 (see Figure 2 and Table B in Appendix), which is comparable to that of the literature [49]. 

Other commonly used measures ask if within the past year, respondents did ever hear a sound (“buzzing, hissing, ringing, humming, roaring, machinery noise”) originating in the ear [49]. It is possible that when tinnitus is infrequent, many people do not report it, because they do not want to identify as someone who hears noises that do not have an external origin. Because our measure allows the option of responding affirmatively to the existence of infrequent tinnitus, it may avoid such identification. Only future research comparing different types of measures could clarify these issues”.

In the subsection “Visual snow and entoptic phenomena”, we expanded the discussion on the possible limitations regarding our measure of entoptic phenomena and how it compares with other measures. As such, we added that “Given the lack of standardized measures to assess entoptic phenomena, we must be cautious regarding the obtained prevalence (see Table C in Appendix). However, we believe that our measure has advantages over others, as we used graphical simulations and resorted to a scale that encompasses a large spectrum of frequency of the experiences (including only occasional experience). These are likely an improvement over measures that only rely on verbal descriptions and less nuanced scales of frequency of occurrence of entoptic phenomena. Only future research using different measures, including in-person interviews might clarify the question of the real prevalence”.

In the subsection “Visual snow and absorption”, we highlighted that the measure of absorption we used is commonly used in surveys, and it yields convergent findings with studies in other contexts. In this regard, we added that “The MODTAS is a commonly used measure of absorption with results confirming its criterion validity [37,54,61-68], including in experimental [62,65,66], and longitudinal studies [63]”.

References cited in this new part (including newly added) are the following:

37. Jamieson GA. The Modified Tellegen Absorption Scale: a clearer window on the structure and meaning of absorption. J Australian Clin Exp Hypn 2005; 33:119-139.

54. Ellero J, Costa RM. Altered states of consciousness, absorption, and sexual responsiveness. Psicol Saúde Doenças 2020; 21: 782-795.

61. Leppanen, M. L., & Kim, K. Absorption and mindfulness reflect distinct patterns of attentional control and self-related processing. J Indiv Diff 2022; 43: 143-151.

62. Timmermann C, Roseman L, Williams L, Erritzoe D, Martial C, Cassol H, Laureys S, Nutt, Carhart-Harris R. DMT models the near-death experience. Front Psychol 2018; 9: 1424.

63. Haijen ECHM, Kaelen M, Roseman L, Timmermann C, Kettner H, Russ S, Nutt D, Daws RE, Hampshire ADG, Lorenz R, Carhart-Harris RL. Predicting Responses to Psychedelics: A Prospective Study. Front Pharmacol 2018; 9:897.

64. Cardeña E, Terhune DB. Hypnotizability, personality traits, and the propensity to experience alterations of consciousness. Psychol Conscious: Theory Res Pract 2014; 1:292-307.

65. Terhune DB, Luke DP, Kaelen M, Bolstridge M, Feilding A, Nutt D, Carhart-Harris R, Ward J. A placebo-controlled investigation of synaesthesia-like experiences under LSD. Neuropsychologia 2016; 88: 28-34.

66. Studerus E, Gamma A, Kometer M, Vollenweider FX. Prediction of psylocibin response in healthy volunteers. PLoS One 2012; 7: e30800.

67. Levine A, Luke D. Apophenia, absorption and anxiety: Evidence for individual differences in positive and negative experiences of Hallucinogen Persisting Perception Disorder. J Psychedelic Stud 2022; 6: 88-103.

68. Wendler E, Schubert E. Synaesthesia, creativity and obsessive-compulsive disorder: Is there a link? Creat Res J 2019; 31: 329-334.

>>>>> On this point, and apologies for introducing another (albeit related) point, to what extent is it possible to differentiate different types of visual snow as entirely separate phenomena based on underlying mechanisms. Based purely on personal experience, I fully recognise the experience of visual snow resulting from sudden changes in blood pressure, but would not relate this to the experience of clinical visual snow, related to cortical excitability. A concern here is that the definition is very broad, allowing for a lot of positive responses.

R: That is an interesting point. In the subsection “Triggers of visual snow”, we added that “Since visual snow has various triggers, it is unclear if it has always the same underlying neural mechanisms, and if these are the same occurring in untriggered visual snow; for example, it is not possible to know if visual cortex hyperexcitability is always present. One way to get a clearer picture may be to apply measures of neural activity like EEG or fMRI and compare the phenomena and their neural correlates under different induction conditions, if this is possible in the lab. This may be combined with the development of a specific questionnaire about visual snow that contains a particular focus on the comparison of different inducing conditions”. 

>>>>> Finally on the point of data availability - although a link has been I could not see the data, I think they have not been uploaded yet.

R: Yes, we have the data ready to upload, but we have not done it yet. We will upload the data after acceptance of the manuscript.

---

## [Editor Report · Decision Letter 2]

18 Oct 2022

Prevalence of visual snow and relation to attentional absorption

PONE-D-21-32390R2

Dear Dr. Costa,

We’re pleased to inform you that your manuscript has been judged scientifically suitable for publication and will be formally accepted for publication once it meets all outstanding technical requirements.

Kind regards,

Guido Hesselmann

Academic Editor

PLOS ONE